# SparseDiff: Sparse Discrete Diffusion for Scalable Graph Generation

**Yiming Qin**                                                      *yiming.qin@epfl.ch*
*Ecole Polytechnique Fédérale de Lausanne (EPFL)*

**Clément Vignac**                                          *vignac@isomorphiclabs.com*
*Ecole Polytechnique Fédérale de Lausanne (EPFL)*

**Pascal Frossard**                                              *pascal.frossard@epfl.ch*
*Ecole Polytechnique Fédérale de Lausanne (EPFL)*

**Reviewed on OpenReview:** *https://openreview.net/forum?id=kuJ3lpxnVC*

## Abstract

Graph generative models encounter significant scaling challenges due to the need to predict the presence or type of edges for every node pair, resulting in quadratic complexity. While some models attempt to support large graph generation, they often impose restrictive assumptions, such as enforcing cluster or hierarchical structures, which can limit generalizability and result in unstable generation quality across various graph types. To address this, we introduce SparseDiff, a novel diffusion framework that leverages the inherent sparsity in large graphs - a highly relaxed assumption that enables efficient sparse modeling without sacrificing generation quality for different datasets. SparseDiff reduces the complexity of the three core components in graph diffusion models. It first introduces a noising trajectory that preserves sparsity with more memory-efficient computation. During training, SparseDiff uses a denoising network based on convolutional attention layers over a sparse edge subsets combining edge-based graph attention and query edge-based random attention mechanisms, maintaining expressiveness with reduced memory usage. Finally, for inference, at each denoising step, SparseDiff generates edge subsets iteratively, progressively reconstructing the adjacency structure. SparseDiff achieves state-of-the-art results on both small and large datasets, showing its robustness across varying graph sizes and its scalability. Additionally, it ensures faster convergence for large graphs, achieving a fourfold speedup on the large-scale Ego dataset compared to dense models. SparseDiff's efficiency, combined with its effective control over space complexity, positions it as a powerful solution for scaling applications involving large graphs. [1]

## 1 Introduction

Graph generation plays a pivotal role in various fields, such as molecular chemistry (Vignac et al., 2023b), neural architecture search (Asthana et al., 2024) and social network analysis (Schweimer et al., 2022), for its ability to model complex relationships and create realistic structured data. Over the past decades, random graph models have played a foundational role in graph generation (Erdős et al., 1960; Barabási, 2013). However, their limitations in capturing complex dependencies in real-world data have shifted research toward machine learning based graph generative models. Traditional frameworks like generative adversarial networks (De Cao & Kipf, 2018) and variational autoencoders (Simonovsky & Komodakis, 2018) have primarily focused on small graphs. Recently, denoising diffusion models (Jo et al., 2022; Niu et al., 2020), especially those employing discrete modeling to better capture graph structure (Vignac et al., 2023a), have emerged, setting

---

[1]Codes available at `https://github.com/qym7/SparseDiff`.

Figure 1: SparseDiff employs three key components for efficiency: (1) a noise model that constructs a sparse noisy graph $G^t$ with high probability, preserving sparsity and enabling more efficient computation, (2) a denoising network trained over all nodes and a sparse subset of query edges with controllable size, $\boldsymbol{E}_q$, and (3) an iterative inference step that, at each time step $t$, progressively fills the adjacency matrix using sparse inputs given by $\boldsymbol{E}_q^i$, for $i \in \{1, 2, 3\}$.

new benchmarks in graph generation tasks and improving scalability by generating graphs with up to 200 nodes. Nonetheless, scaling to even larger graphs remains challenging, restricting application in fields such as protein generation (Yim et al., 2023), histopathology (Madeira et al., 2023), transportation (Rong et al., 2023) or anomaly detection in financial systems (Li et al., 2023).

Motivated by these limitations and recognizing that most real-world graphs are inherently sparse — for instance, cells in digital pathology connect only with immediate neighbors (Madeira et al., 2023), and connections in social networks are sparse (McCallum et al., 2000) — we propose SparseDiff. SparseDiff is a discrete diffusion framework that leverages graph sparsity without additional assumptions. Our technical contributions are fourfold. (1) We observe the universal sparsity of real-world graphs, adopting a sparse triplet graph representation $(\boldsymbol{E}, \boldsymbol{X}, \boldsymbol{Y})$ in place of dense formats, which scales more efficiently for graphs with different sizes (see Appendix E.2). This enables the realization of the remaining contributions of SparseDiff, representing the three core components of diffusion models (detailed in Section 2.1) as illustrated in Figure 1: (2) The *efficient noise model* (Section 3.1) introduces a sparse noising process that preserves the sparsity of graphs along the forward diffusion trajectory with more space-efficient computation. (3) The *sparse denoising neural network* (Section 3.2) uses convolutional attention layers adapted to our sparse graph representation. Its complexity scales with the number of non-zero entries of its sparse attention map, which covers existing edges and a tunable number of random node pairs, enabling training on large graphs under controllable space complexity without compromising performance. The model is trained on random query edges at each step, analogous to mini-batches in SGD, allowing optimization under constrained computation resources. (4) The *iterative inference* procedure (Section 3.3) modifies the standard sampling strategy to align with the sparse attention used during training. At each diffusion step, it incrementally reconstructs the dense adjacency matrix by adding edge subsets defined by query edges, maintaining structural consistency and improving efficiency at generation time.

Our experiments demonstrate that SparseDiff consistently achieves state-of-the-art performance on both small graphs, including those with complex priors like molecular graphs, and large graph datasets. Notably, SparseDiff handles much larger graphs (up to 2485 nodes), while the performance of DiGress degrades on graphs with over 200 nodes. Specifically, SparseDiff outperforms both dense models like SPECTRE (Martinkus et al., 2022) and DiGress (Vignac et al., 2023a), as well as other scalable models like EDGE (Chen et al., 2023) and HGGT (Jang et al., 2023). SparseDiff also converges four times faster than dense diffusion models on large graphs, such as social networks. These results empirically demonstrate the effectiveness of

our sparse modeling approach and confirm its robustness through consistently high performance across a diverse range of datasets.

In summary, SparseDiff is able to match the performance of dense diffusion models without restrictive assumptions, pioneering the use of sparse representations in graph diffusion, and distinguishing it from hierarchical and autoregressive methods. SparseDiff is implemented within the discrete diffusion framework, specifically following the setup of Vignac et al. (2023a). Its noise model is particularly well-suited for scenarios where non-existent edges dominate, such as marginal trajectories in SparseDiff or absorbing trajectories that converge to empty graphs. Additionally, the sparse attention mechanism and iterative sampling are broadly applicable to both diffusion or flow-based generative models, as they focus solely on sparsifying the attention map and the corresponding adjacency matrix. As a remark, while SparseDiff can potentially improve diffusion and flow models with sparse noisy graphs, it provides no benefit for continuous models that do not preserve sparsity, as they must be modeled using a dense adjacency matrix.

## 2  Related Work

### 2.1  Denoising Diffusion Models for Graphs

Diffusion models have gained increasing popularity due to their impressive performance across various generative tasks in fields such as computer vision (Dhariwal & Nichol, 2021; Ho et al., 2022), protein generation (Baek et al., 2021; Ingraham et al., 2022), and audio synthesis (Kong et al., 2020). These models are characterized by three core components. The first is a Markovian noise model, which progressively corrupts a data point $x$ to a noisy sample $z^t$ over iterative steps $t$ from 1 to $T$, until it conforms to a simple predefined prior distribution at step $T$. The second component is a denoising network, parametrized by $\theta$, which is trained to restore the corrupted data back to its less noisy state. Typically, this network aims to predict the original data $x$ given the noisy sample $z^t$. The third component is the reverse process for data generation, where a fully noisy data point $z^T$ is first sampled from a prior distribution. The denoising network then operates at each time step $t \in [T, ..., 1]$ to predict the less noisy distribution according to $p_\theta(z^{t-1}|z^t) = \int_x q(z^{t-1}|z^t, x) \, dp_\theta(x|z^t)$, from which a new data point $z^{t-1}$ is sampled. While this integral is generally difficult to evaluate, two prominent frameworks, Gaussian diffusion (Ho et al., 2020) and discrete diffusion (Austin et al., 2021), facilitate its efficient computation.

Initial graph diffusion models employed Gaussian noise directly on adjacency matrices (Niu et al., 2020; Jo et al., 2022). These models use a graph attention network to regress the added noise $\epsilon$, where $\epsilon = z^t - x$, effectively regressing the noise up to an affine transformation, akin to regressing the discrete clean graph. To preserve the inherent discreteness of graphical data, subsequent models (Vignac et al., 2023a; Haefeli et al., 2022) have leveraged discrete diffusion, reformulating graph generation as a series of classification tasks, and achieving top-tier results. However, these models require predictions for all pairs of nodes, which implies a quadratic space complexity and thus restricts their scalability. We further provide a detailed discussion of discrete graph diffusion models in Appendix E.2.

### 2.2  Scalable Graph Generation

Efforts to enhance the scalability of graph generative models mainly follow two paradigms: hierarchical refinement and subgraph aggregation. The hierarchical refinement approach initially generates a low-resolution graph, which undergoes successive refinements for enhanced detail (Yang et al., 2021; Karami, 2023). For instance, the HGGT model (Jang et al., 2023) employs a hierarchical $K^2$-tree representation. For molecule generation, fragment-based models (Jin et al., 2018; 2020; Maziarz et al., 2022) adeptly assemble compounds using pre-defined molecular fragments. Recently, Bergmeister et al. (2023) proposes a hierarchical diffusion model based on spectrum-preserving local expansion algorithms, enabling the generation of non-attributed large graphs. On the other side, the subgraph aggregation approach divides larger graphs into smaller subgraphs, which are subsequently combined. For instance, SnapButton (Yang et al., 2021) enhances autoregressive models (Liu et al., 2018; Liao et al., 2019; Mercado et al., 2021) by merging subgraphs, and SaGess (Limnios et al., 2023) trains a dense diffusion model to generate subgraphs sampled from a large graph that are then merged.

Additionally, some approaches predict all node pairs in an auto-regressive manner. Kong et al. (2023) integrated diffusion with autoregressive models, suggesting learning the node ordering, which is theoretically as difficult as isomorphism testing. Alternatively, EDGE (Chen et al., 2023) uses absorbing states to create sparse diffusion by first generating a node degree distribution $\boldsymbol{d}^0$ and gradually constructing the adjacency matrix $\boldsymbol{A}$ based on node degree changes during inference. While this factorization is universally applicable, the conditional distribution $p_\theta(\boldsymbol{A}|\boldsymbol{d}^0)$ may occasionally lead to degree distributions that are not feasible for undirected graphs during sampling, introducing a slight misalignment between training and generation. EDGE's performance also falls behind SparseDiff on both small and large graphs. Besides, while latent diffusion is commonly used for scalability in vision tasks, applying it to graphs is more challenging due to permutation equivariance, where the decoded graph can appear in any permutation, necessitating graph matching. Current latent graph diffusion models either use predefined orderings (Evdaimon et al., 2024), or perform diffusion on node features followed by link prediction (Yang et al., 2024), with the latter underperforming SparseDiff or other dense models (see App. E.1 for details).

Overall, scalable generation models often introduce dependencies on node orderings, incorporate assumptions about data distributions, or leverage specific graph properties, while latent graph diffusion faces challenges for graph matching. In contrast, the SparseDiff model described in the next section aims at making no assumption besides graph sparsity, offering a more general and streamlined framework for graph generation, while preserving competitive performance across a wide range of graphs with different sizes.

## 3 SparseDiff: Sparse Discrete Diffusion for Graph Generation

We now introduce SparseDiff, a Sparse Denoising Diffusion Model that matches the performance of dense models while significantly enhancing the scalability of graph diffusion to graphs with over $2,000$ nodes, a significant improvement over previous dense models limited to around 200 nodes.

Unlike previous graph diffusion models, SparseDiff builds on a sparse representation of graphs. A graph $G$, consisting of $n$ nodes and $m$ edges, is represented as a triplet $(\boldsymbol{E}, \boldsymbol{X}, \boldsymbol{Y})$. Here, $\boldsymbol{E} \in \mathbb{N}^{2 \times m}$ indicates the edge list with the indices of endpoints. Node and edge attributes are considered to be discrete and are encoded in a one-hot format as $\boldsymbol{X} \in \{0,1\}^{n \times a}$ and $\boldsymbol{Y} \in \{0,1\}^{m \times b}$, where $a$ and $b$ are the number of classes, respectively. In particular, non-existing edges are considered an additional edge type, while edges in $\boldsymbol{E}$ are referred to as *existing edges*. This sparse representation is widely supported by standard graph processing packages such as Pytorch Geometric (Fey & Lenssen, 2019).

This work focuses solely on undirected graphs with discrete attributes, although continuous labels can be integrated, in a similar way to Vignac et al. (2023b). All considered graphs are free of self-loops. Figure 1 provides an overview of SparseDiff framework. Further details on the training and sampling processes are provided in Algorithms 1 and 2, respectively. In the following parts, we specifically focus on three critical components of the diffusion model as detailed in Section 2.1: the noise model, the denoising network, and the sampling algorithm. Appendix A summarizes the notation used throughout the paper, and Appendix A.1 further provides a detailed breakdown of the space efficiency for each component.

### 3.1 Efficient Noise Model

To improve memory efficiency, SparseDiff employs a dedicated noise model that maintains a similar sparsity level of the noisy graph $G^t$ throughout the noising trajectory under theoretical guarantee, as detailed in Section 3.1.1. This model also reduces the space complexity for computing noisy graph distribution as detailed in Section 3.1.2.

#### 3.1.1 Sparse Trajectory

Given that Gaussian noise applied to the adjacency matrix during diffusion typically results in dense noisy graphs, where all edge entries in the adjacency matrix acquire continuous values (Niu et al., 2020; Jo et al., 2022) — we opt for a discrete diffusion framework. In this framework, we sample a graph structure from the noisy distribution, allowing us to focus only on existing edges within the noisy graph, thereby reducing the number of edges necessary for computation. In the discrete graph diffusion model, the noisy trajectory at

each step is defined by $q(G^t|G^{t-1}) = (\boldsymbol{X}\boldsymbol{Q_X^t}, \boldsymbol{Y}\boldsymbol{Q_Y^t})$, where $\boldsymbol{Q}^t$ represents the Markov transition matrix for that step $t$, which transforms $G^{t-1}$ into a noisier distribution until reaching $G^T$, which follows a predefined prior distribution that is easy to sample. Different types of Markov transition matrices are employed to corrupt the graphs into various prior distributions, including uniform distributions, a special absorbing state (Austin et al., 2021), or marginal distributions (Vignac et al., 2023a).

In this work, we employ the marginal transition model, which favors transitions towards the dominant class in the marginal distribution of the data. This strategy is particularly effective for preserving graph sparsity by naturally biasing toward non-existing edges, which is the dominant edge class for large graphs. Formally, considering the marginal distribution vectors $\boldsymbol{p_X}$ for node types and $\boldsymbol{p_E}$ for edge types, and denoting their transposes by $\boldsymbol{p}'$, the noise level at each step $t$ is regulated by $\beta^t$, with $\alpha^t = 1 - \beta^t$. Formally, the marginal transition matrices are defined as follows:

$$\boldsymbol{Q_X^t} = \alpha^t \boldsymbol{I} + \beta^t \mathbf{1}_a \boldsymbol{p_X'}, \quad \boldsymbol{Q_Y^t} = \alpha^t \boldsymbol{I} + \beta^t \mathbf{1}_b \boldsymbol{p_Y'}.$$

Here, $\mathbf{1}_a$ and $\mathbf{1}_b$ are column vectors of ones with dimensions equal to the number of classes $a$ for nodes and $b$ for edges. These matrices incorporate a first term, the identity matrix $\boldsymbol{I}$, to preserve the distribution from $G^{t-1}$, and a second term to introduce noise aligned with the marginal distributions, namely $\boldsymbol{p_X}$ and $\boldsymbol{p_E}$.

By employing continuous multiplication, we can derive the distribution at step $t$ directly from the initial clean graph using $q(G^t|G) = (\boldsymbol{X}\bar{\boldsymbol{Q}}_{\boldsymbol{X}}^t, \boldsymbol{Y}\bar{\boldsymbol{Q}}_{\boldsymbol{Y}}^t)$, facilitating an immediate transition to the noisy state without the need for iterative step-by-step calculations. For instance, the cumulative transition matrix $\bar{\boldsymbol{Q}}^t$ for nodes $\boldsymbol{X}$ is as follows: $\bar{\boldsymbol{Q}}_{\boldsymbol{X}}^t = \boldsymbol{Q}_{\boldsymbol{X}}^1 \boldsymbol{Q}_{\boldsymbol{X}}^2 \ldots \boldsymbol{Q}_{\boldsymbol{X}}^t = \bar{\alpha}^t \boldsymbol{I} + (1 - \bar{\alpha}^t) \mathbf{1}_a \boldsymbol{p_X'}$, where $\bar{\alpha}^t = \alpha^1 \alpha^2 \ldots \alpha^t$. The parameter $\bar{\alpha}^t$ starts very close to 1 at $\bar{\alpha}^1$ and approaches 0 at $\bar{\alpha}^T$, reflecting a gradual increase in noise influence over the diffusion process.

We note that this choice of marginal noise model does not guarantee that the noisy graph is always sparse. However, it is the case with high probability, as stated by the following lemma, which is an application of Desolneux et al. (2008) (detailed in Appendix A.2).

**Lemma 3.1.** *(Probability Bound for Sparsity in Noisy Graphs) Consider an undirected graph with $n$ nodes, $m$ edges, and no self-loops. If the edge ratio given by $m/\binom{n}{2}$ is denoted as $r$, and the edge ratio in the noisy graph sampled from the marginal transition noise model is given by $r_t$, then for $n$ sufficiently large and $r < \frac{1}{4}$, for any $r < k < 1$, we have:*

$$\log(\mathrm{P}[r_t \geq k]) \sim -\binom{n}{2}\left(k \log \frac{k}{r} + (1-r)\log\frac{1-k}{1-r}\right) \tag{1}$$

This lemma demonstrates that, in large and sparse graphs, the probability of the edge ratio $r_t$ in the noisy graph exceeding a threshold $k$ (where $k > r$) declines exponentially with graph size. For instance, in graphs with a low edge ratio $r$ and setting the edge threshold at $k = 2r$, the probability that the noisy graph exceeds $k$ edges is approximately $c_1 e^{-c_2 n^2 r}$, where $c_1$ and $c_2$ are constants. This probability decreases substantially as the graph size $n$ grows.

### 3.1.2 Sparse Computation

The second requirement for the noise model is to reduce the space complexity for computing the noisy distribution. Standard discrete diffusion models represent all edges using $\boldsymbol{Y} \in \mathbb{R}^{n \times n \times b}$, with transition probabilities computed as $\boldsymbol{Y}\bar{\boldsymbol{Q}}_{\boldsymbol{Y}}^t \in \mathbb{R}^{n \times n \times b}$, resulting in $\mathcal{O}(n^2)$ space complexity. To address this, we separate existing edges from non-existing ones, as the latter primarily contribute to quadratic complexity in sparse graphs. We define the set of indices of non-existing edges as $\boldsymbol{E}_{ne} \in \mathbb{N}^{2 \times (\binom{n}{2} - m)}$, where $\boldsymbol{E} \in \mathbb{N}^{2 \times m}$ represents the set of indices of existing edges. Together, $\boldsymbol{E}$ and $\boldsymbol{E}_{ne}$ form a partition of all node pairs.

Specifically, we compute $\boldsymbol{Y}\bar{\boldsymbol{Q}}_{\boldsymbol{Y}}^t \in \mathbb{R}^{m \times b}$ for existing edges with space complexity $\mathcal{O}(m)$, we then sample new edge labels directly from this distribution, while removing edges transitioning to non-existing states to maintain sparsity in $G^t$. For non-existing edges $\boldsymbol{E}_{ne}$, which typically drive quadratic complexity, we introduce a novel three-step approach to sample efficiently without dense adjacency matrices.

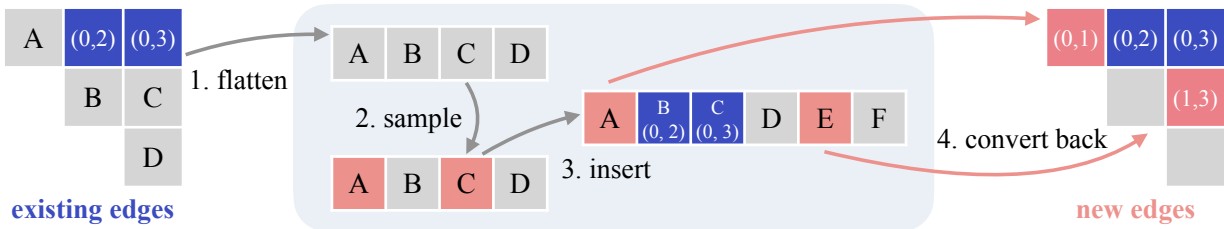

Figure 2: Efficient sampling of new edge positions among non-existing positions is achieved using only the edge list, avoiding the quadratic space complexity of an adjacency matrix. The process involves: (1) Conceptually **flattening** the pair representations into a linear array, (2) **Sampling** uniformly 2 positions from 4 non-occupied positions from $E_{ne}$, selecting A (1st) and C (3rd), (3) **Inserting** offsets of 0 and 2 to the positions of A and C to account for the count of existing edges before the selected positions, resulting in positions A (1st) and E (5th), and (4) **Converting** these positions back to index pairs $(0, 1)$ and $(1, 3)$.

1. **Sampling new edge count:** Let $\bar{m}_t = \binom{n}{2} - m_t$ denote the number of non-existing edges in $G^t$. The number of new edges is sampled as $k \sim \mathcal{B}(\bar{m}_t, q_t)$, where $q_t = 1 - \boldsymbol{Q}^t[0, 0]$ represents the probability of a non-existing edge transitioning to an existing one.

2. **Sampling positions:** The new edge positions are selected uniformly at random from $E_{ne}$.

3. **Sampling edge attributes:** The attribute of each new edge is drawn from the Multinomial distribution Multinomial($\boldsymbol{Q}^t[0, 1 : b]$), representing the transition probabilities over existing edge types.

In this approach, Step 1 employs a Binomial distribution $\mathcal{B}(\bar{m}_t, q_t)$ to efficiently disregard edges that remain non-existent, focusing computation solely on newly emerging edges to reduce overhead. Step 2, sampling new edge positions, presents additional challenges, as it requires efficiently sampling a certain number of edges from a batch of graphs with varying sizes. Moreover, the sampling must only occur from non-occupied positions, and we must only rely on the edge list rather than the adjacency matrix to avoid high space complexity. As illustrated in Figure 2, we propose an algorithm to efficiently handle this sparse sampling. The method begins by conceptually flattening all vacant positions. Simple random sampling is then performed on these positions using `randint` to select 2 elements from 4. Crucially, after sampling, the indices have to be adjusted by adding offsets to account for existing edges through a specially designed algorithm, before remapping them back to index pairs. Finally, as the noisy distribution for non-existing edges is uniform, i.e., $\boldsymbol{Q}^t[0, 1 : b]$, Step 3 samples directly from a fixed distribution, avoiding redundant matrix multiplications required in prior models.

## 3.2 Efficient Denoising Neural Network

The traditional model for graph diffusion typically outputs the probabilities for all edges and nodes, which aligns closely with attention layers. These layers explicitly encode edge features with a computational complexity of $O(ln^2 d_e)$, where $l$ is the number of layers and $d_e$ represents the edge feature dimension. In contrast, message-passing methods only account for existing edges, significantly reducing the complexity to $O(lmd_e)$, where $m$ is the number of existing edges.

However, as discussed in Appendix D.5.1, relying solely on message-passing and link prediction using node features tends to perform worse on both small and large datasets. This may be due to the inability to capture long-term and complex interactions between distant nodes, resulting in degraded performance in practice. To address this, we propose bridging the gap between a fully connected attention layer and message-passing with a random attention mechanism. Specifically, we first integrate a convolutional transformer layer (Shi et al., 2020) that restricts attention to existing edges. Additionally, we incorporate random attention by sampling query edges randomly from all possible node pairs. By expectation, the model learns the full attention map between nodes, enabling it to capture more complex interactions.

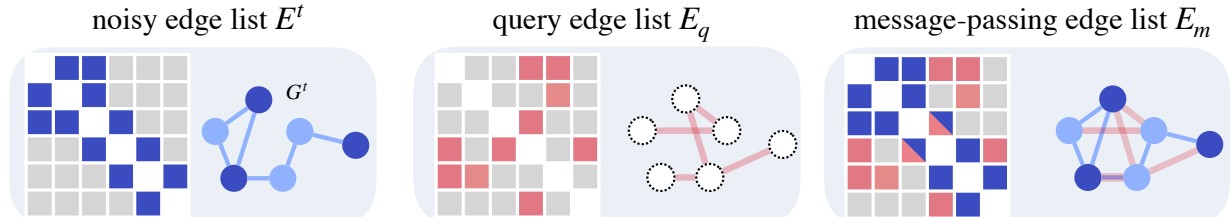

Figure 3: The sparse attention map used in the sparse denoising network is based on the message-passing edge list $\boldsymbol{E}_m$. This edge list consists of noisy edges for graph attention $\boldsymbol{E}^t$ (shown in blue) and uniformly sampled query edges for random attention $\boldsymbol{E}_q$ (shown in red).

### 3.2.1  Edge Prediction Module using Sparse Attention

As previously discussed, relying solely on message-passing layers for edge prediction yields limited results. To enhance performance, we propose a random attention mechanism through a randomly selected subset of node pairs, referred to as query edges. As shown in Figure 3, the input to the denoising network, referred to as message-passing edges, is denoted by their indices and categories $(\boldsymbol{E}_m, \boldsymbol{Y}_m)$. This includes two edge sets: (1) noisy edges $(\boldsymbol{E}^t, \boldsymbol{Y}^t)$, representing existing edges in the noisy graph $G^t$, and (2) query edges $(\boldsymbol{E}_q, \boldsymbol{Y}_q)$, which are randomly selected for attention and loss computation. Formally, the model takes the following edges as input:

$$\boldsymbol{E}_m = \boldsymbol{E}^t \cup \boldsymbol{E}_q, \quad \boldsymbol{Y}_m = \boldsymbol{Y}^t \cup \boldsymbol{Y}_q. \tag{2}$$

Noisy edges indexed by $\boldsymbol{E}^t$ preserve the topological information of the noisy graph, while query edges indexed by $\boldsymbol{E}_q$ are uniformly sampled from all node pairs, covering both existing and non-existing edges. This sampling strategy enables the model to predict arbitrary edge types while ensuring an unbiased loss estimator relative to dense diffusion models.

Additionally, the query edges in $\boldsymbol{E}_q$ facilitate random attention and serve as implicit shortcuts in the message-passing network, enabling graph rewiring. As noted in studies such as (Alon & Yahav, 2020; Topping et al., 2021; Di Giovanni et al., 2023), this mechanism enhances information propagation and helps mitigate over-squashing issues by introducing additional rewiring edges denoted by $\boldsymbol{E}_r = \boldsymbol{E}_q \setminus \boldsymbol{E}^t$.

To control the number of query edges, we define the sparsity parameter $\lambda$ as $\lambda = \frac{|\boldsymbol{E}_q|}{n^2}$. Given that query edges may overlap with noisy edges and that the number of noisy edges $m_t$ approximates the number of edges $m$ in the clean graph (according to Lemma 3.1), the total number of message-passing edges satisfies $|\boldsymbol{E}_m| \leq |\boldsymbol{E}^t| + |\boldsymbol{E}_q| \leq m + \lceil \lambda n^2 \rceil$. Setting $\lambda = \frac{m}{n^2}$ ensures that the computational complexity of SparseDiff remains approximately $\mathcal{O}(|\boldsymbol{E}_m|) = \mathcal{O}(m)$, aligning with the number of edges in the clean graph $G$. In our experiments, $\lambda$ is selected to balance computational efficiency and effective batch size, as detailed in Appendix C.2. This flexibility enables robust performance across different settings, as demonstrated in Table 5.

### 3.2.2  Model Training

Our sparse denoising network adopts a graph transformer architecture featuring normalization, feed-forward, and attention layers (Veličković et al., 2017). It incorporates a sparse attention mechanism for handling sparse data (Shi et al., 2020), and integrates advanced features such as PNA pooling layers (Corso et al., 2020) and FiLM layers (Perez et al., 2018), which are designed to enhance predictive accuracy and effectively manage computational complexity. A detailed discussion of the model architecture is provided in Appendix B.

Training of the network $\phi_\theta$ parameterized by $\theta$ involves predicting query edges $\boldsymbol{E}_q$, and the loss is minimized using the cross-entropy (CE) loss between the predicted distribution $\hat{\boldsymbol{P}}_q^G = (\hat{\boldsymbol{P}}^X, \hat{\boldsymbol{P}}_q^Y)$ and the clean graph $G$. As detailed in Alg. 1, the loss function is computed as follows:

$$\sum_{1 \leq i \leq n} \mathrm{CE}(\boldsymbol{X}_i, \hat{\boldsymbol{P}}_i^X) + \frac{c}{\lambda} \sum_{(i,j) \in \boldsymbol{E}_q} \mathrm{CE}(\boldsymbol{Y}_{ij}, \hat{\boldsymbol{P}}_{ij}^Y), \tag{3}$$

---

**Algorithm 1** Sparse training at step $t$ with the sparsity parameter $\lambda$ (Section 3.1 & 3.2)

---

1: Given the clean graph $G = (\boldsymbol{E}^0, \boldsymbol{X}^0, \boldsymbol{Y}^0)$;
2: Sample the noisy graph $G^t = (\boldsymbol{E}^t, \boldsymbol{X}^t, \boldsymbol{Y}^t)$;
3: Sample query edges $\boldsymbol{E}_q$ of size $\lceil \lambda n^2 \rceil$ ;
4: $\boldsymbol{E}_m \leftarrow \boldsymbol{E}^t \cup \boldsymbol{E}_q, \boldsymbol{Y}_m \leftarrow \boldsymbol{Y}^t \cup \boldsymbol{Y}_q$;          ▷ Construct message-passing edges
5: $G_m \leftarrow (\boldsymbol{E}_m, \boldsymbol{X}^t, \boldsymbol{Y}_m)$;          ▷ Construct the message-passing graph
6: $(\hat{\boldsymbol{P}}^{\boldsymbol{X}}, \hat{\boldsymbol{P}}_q^{\boldsymbol{Y}}) = \phi_\theta(G_m, \boldsymbol{E}_q)$;       ▷ Predict the distribution of nodes and query edges
7: optimizer. step($\mathrm{CE}(\hat{\boldsymbol{X}}^0, \hat{\boldsymbol{P}}^{\boldsymbol{X}}) + \mathrm{CE}(\boldsymbol{Y}_q^0, \hat{\boldsymbol{P}}_q^{\boldsymbol{Y}})$);          ▷ Loss calculation

---

**Algorithm 2** Iterative inference at step $t$ with the sparsity parameter $\lambda$ (Section 3.3)

---

1: Initialize an empty graph $G^{t-1}$ with unlabeled nodes $\boldsymbol{X}^{t-1}$ and no edges;
2: Randomly divide all node pairs into $K = \lceil \frac{1}{\lambda} \rceil$ equal-sized chunks $\{C_0, \cdots, C_{K-1}\}$;
3: **for** k in range($K$) **do**
4:   $\boldsymbol{E}_q \leftarrow C_k$;          ▷ Set query edges
5:   $\boldsymbol{E}_m \leftarrow \boldsymbol{E}^t \cup \boldsymbol{E}_q$;      ▷ Construct message-passing edges and its attributes $\boldsymbol{Y}_m$
6:   $G_m \leftarrow (\boldsymbol{E}_m, \boldsymbol{X}^t, \boldsymbol{Y}_m)$;      ▷ Construct the message-passing graph
7:   $(\hat{\boldsymbol{P}}^{\boldsymbol{X}}, \hat{\boldsymbol{P}}_q^{\boldsymbol{Y}}) = \phi_\theta(G_m, \boldsymbol{E}_q)$;    ▷ Predict the distribution of nodes and query edges
8:   $\hat{\boldsymbol{X}} = \mathrm{Multinomial}(\hat{\boldsymbol{P}}^{\boldsymbol{X}}), \hat{\boldsymbol{Y}}_q = \mathrm{Multinomial}(\hat{\boldsymbol{P}}_q^{\boldsymbol{Y}})$;      ▷ Sample labels
9:   $\boldsymbol{X}^{t-1} \leftarrow \hat{\boldsymbol{X}}$;        ▷ Assign node new labels
10:   $\boldsymbol{Y}^{t-1} \leftarrow \boldsymbol{Y}^{t-1} \cup \hat{\boldsymbol{Y}}_q[\hat{\boldsymbol{Y}}_q! = 0], \boldsymbol{E}^{t-1} \leftarrow \boldsymbol{E}^{t-1} \cup \hat{\boldsymbol{E}}_q[\hat{\boldsymbol{Y}}_q! = 0]$;      ▷ Add existing edges

---

Here, the constant $c$ weights nodes and edges in the loss calculation. It is rescaled by dividing by $\lambda$ to maintain a consistent edge-to-node weight ratio across different $\lambda$ values. Conceptually, training SparseDiff is analogous to using stochastic gradient descent instead of standard gradient descent. Query edges act as random mini-batches, so SparseDiff's updates remain aligned with those of dense diffusion models.

### 3.3 Iterative Inference

SparseDiff also remains memory-efficient during the inference stage, as visualized in Part (3) of Figure 1. We start by sampling the number of nodes $n$ of the generated graph from the node distribution of the training set, which remains constant during the reverse process. Next, we sample a random graph from the prior distribution $G^T \sim \prod_{i=1}^n \mathrm{Cate}(\boldsymbol{p_X}) \times \prod_{1 \le i < j \le n} \mathrm{Cate}(\boldsymbol{p_Y})$, where $\boldsymbol{p_X}$ and $\boldsymbol{p_Y}$ represent the marginal probabilities of node and edge classes, respectively. The categorical distribution $\mathrm{Cate}(\boldsymbol{p})$ is used for both nodes and edges. The sparse denoising network $\phi_\theta$ is then recursively applied to predict the clean graph from the noisy one. The denoising processes of SparseDiff are further visualized in Figure 7.

Directly predicting the entire graph at each diffusion step $t$ is impractical due to quadratic memory requirements. Moreover, using dense graphs during inference could lead to a distribution shift compared to the training stage, due to changes in the number of edges used for message-passing. To mitigate this, we implement an iterative procedure to progressively cover all node pairs in $G^{t-1}$. As detailed in Algorithm 2 and in Part (3) of Figure 1, we divide all node pairs randomly into $K = \lceil \frac{1}{\lambda} \rceil$ equally-sized sets, representing the query edges for each prediction step[2]. During each iteration $k$, the noisy graph $G^t$ remains identical, and a message-passing edge list $\boldsymbol{E}_m$ is constructed using noisy edges and query edges from the $k^{th}$ set denoted by $\boldsymbol{E}_q^k$. SparseDiff then predicts the distributions for these query edges, samples labels, and integrates edges classified as existing into $G^{t-1}$.

---

[2]When $\frac{n(n-1)}{2}$ is not divisible by $K$, we adjust by slightly overlapping the last set with the previous one.

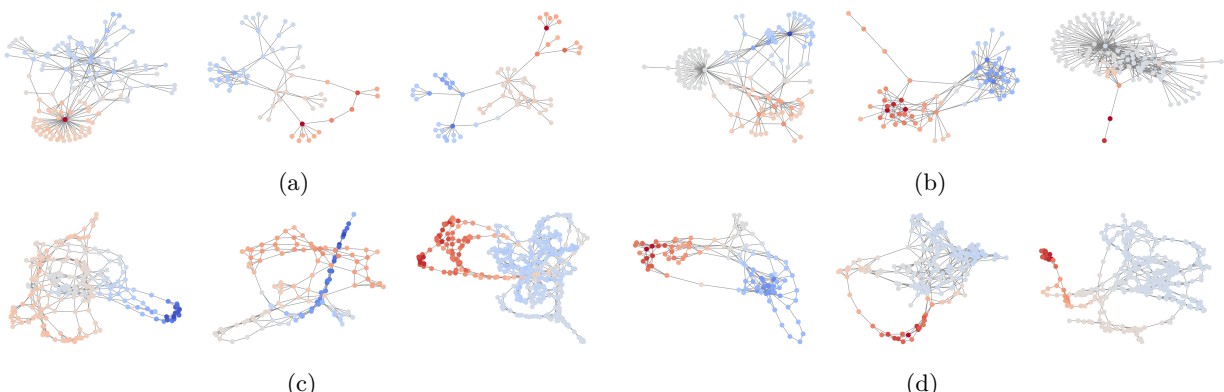

Figure 4: Samples from SparseDiff trained on large graphs. (a) Ego training set (50 to 399 nodes); (b) Generated Ego graphs; (c) Protein training set (100 to 500 nodes); (d) Generated Protein graphs.

This iterative approach allows SparseDiff to maintain favorable memory complexity, albeit at the cost of increased sampling time due to iterations at each diffusion step. However, unlike many scalable models, SparseDiff does not impose additional assumptions about data distribution, such as clustering or degree distribution. Despite the increased sampling time, the generation time for SparseDiff remains efficient due to its ability to utilize larger batch sizes and to accelerate model computations using sparse inputs, as reported in Table 15 of Appendix D.4.

Finally, drawing inspiration from D3PM (Austin et al., 2021) and DDIM (Song et al., 2020), we propose a method to accelerate inference by reducing the inference steps by a factor $k$. In particular, at each step $t$, the model predicts $q(G^{t-k}|G^t, G) \propto G^t(Q^t)' \odot G\bar{Q}^{t-k}$ instead of $q(G^{t-1}|G^t, G) \propto G^t(Q^t)' \odot G\bar{Q}^{t-1}$. The results for acceleration are reported in Table 7.

## 4 Experiments

We evaluate SparseDiff on diverse graph datasets against a broad range of baselines, including GraphRNN (You et al., 2018), GRAN (Liao et al., 2019), GraphNVP (Madhawa et al., 2019), SPEC-TRE (Martinkus et al., 2022), GDSS (Jo et al., 2022), DiGress (Vignac et al., 2023a), DruM (Jo et al., 2023), and scalable models such as BiGG (Dai et al., 2020), GraphARM (Kong et al., 2023), EDGE (Chen et al., 2023), HiGen (Karami, 2023), and HGGT (Jang et al., 2023). We refer to the method from Bergmeister et al. (2023) as GraphLE. We report results as originally published to ensure fair and consistent comparison. DiGress (Vignac et al., 2023a), as the primary baseline, is reproduced using SparseDiff's settings due to missing checkpoints. Training and architectural hyperparameters are detailed in App. D.6.

SparseDiff metrics are reported as mean ± standard deviation, derived from five samples, mitigating variance of inference. We **bold** the best-performing method for each metric. The results underscore SparseDiff's significant competitive advantage on datasets containing larger graphs, such as Planar and SBM (Martinkus et al., 2022), Protein (Dobson & Doig, 2003), Ego and CORA (Sen et al., 2008), and Facebook (Mcauley & Leskovec, 2014) alongside its state-of-the-art performance on datasets with small molecules, including QM9 (Wu et al., 2018) and MOSES (Polykovskiy et al., 2020). Notably, SparseDiff is the only model that demonstrates competitive performance across both large and small graphs, handling both attributed and unattributed graphs effectively.

### 4.1 Large Graph Generation

**Dataset** We evaluate SparseDiff on diverse graph datasets to demonstrate its scalability and versatility. First, we test its ability to generate edge-crossing-free planar graphs with 64 nodes. Next, we assess its capacity to generate graphs with 2 to 5 communities using Stochastic Block Model (SBM) graphs, scaling up to 200 nodes — the largest size seen in models like DiGress. We also evaluate Ego and Protein datasets, with graphs up to 500 nodes, representing citation relationships and amino acid interactions within 6 Angstroms. The largest edge ratio for these datasets is 8.8%, confirming their sparsity. Detailed dataset statistics are in

Table 1: Sample quality on large graphs. The mean ratios to the reference of the Degree, Cluster, Orbit, and Spectre MMD metrics are reported to enable a comprehensive comparison.

| Class | Model | Degree($10^{-3}$)↓ | Cluster ($10^{-2}$)↓ | Orbit ($10^{-2}$)↓ | Spectre ($10^{-3}$)↓ | $\overline{\text{Ratio}}$ ↓ | RBF ($10^{-2}$)↓ |
|---|---|---|---|---|---|---|---|
| *Protein* | *Reference* | 0.3 | 0.7 | 0.3 | 0.5 | 1.0 | 1.4 |
| Dense | GRAN | 2.0 | 4.9 | 13 | 5.1 | 17 | – |
|  | DiGress | 5.9±0.1 | 10±1.4 | 5.1±1.8 | 2.9±0.5 | 14±2.3 | 7.1±1.5 |
| Sparse | DruM | 1.9 | 6.6 | 3.5 | 3.0 | 8.4 | – |
|  | BiGG | **1.0** | **2.6** | 2.3 | 4.5 | 5.9 | – |
|  | HiGen | 1.2 | 4.4 | 2.3 | 2.5 | 5.7 | – |
|  | GraphLE | 3.0 | 3.1 | **0.5** | **1.3** | 4.7 | – |
|  | SparseDiff | 1.5±0.3 | 3.4±0.3 | **0.5**±0.8 | 1.4±0.2 | **3.6**±1.1 | **3.8**±0.7 |
| *Ego* | *Reference* | 0.2 | 0.7 | 0.7 | 1.0 | 1.0 | 0.9 |
| Dense | DiGress | 8.9±1.6 | 5.4±0.4 | 3.0±0.3 | 19±3.2 | 19±3.1 | 3.4±0.8 |
| Sparse | EDGE | 58 | 18 | 5.2 | – | 107 | 6.6 |
|  | HiGen | 47 | **0.3** | 3.9 | – | 81 | 4.5 |
|  | SparseDiff | **3.7**±0.4 | 3.2±0.1 | **2.0**±0.4 | **5.6**±0.8 | **7.9**±0.9 | **2.6**±0.3 |

Table 2: Sample quality on synthetic graphs. The mean ratios to the reference of the Degree, Cluster and Orbit MMD metrics are reported to enable a comprehensive comparison.

| Dataset | Stochastic block model | | | | | Planar | | | | |
|---|---|---|---|---|---|---|---|---|---|---|
| Model | Degree↓ | Cluster↓ | Orbit↓ | $\overline{\text{Ratio}}$ ↓ | V.U.N.↑ | Degree ↓ | Cluster ↓ | Orbit↓ | $\overline{\text{Ratio}}$ ↓ | V.U.N.↑ |
| *Reference* | 0.9 | 3.3 | 2.6 | 1.0 | 100% | 0.2 | 3.1 | 0.1 | 1.0 | 100% |
| GRAN | 5.5 | 5.8 | 7.9 | 3.6 | 25% | 0.7 | 4.3 | **0.1** | 2.0 | 0% |
| GG-GAN | 3.5 | 7.0 | 5.9 | 2.8 | 25% | 63 | 118 | 123 | 528 | 0% |
| SPECTRE | 1.5 | 5.2 | 4.1 | 1.6 | 53% | 0.5 | 7.9 | **0.1** | 2.0 | 25% |
| DruM | **0.7** | **4.9** | 4.5 | 3.6 | **85**% | 0.5 | 3.5 | **0.1** | 1.5 | 90% |
| HiGen | 5.5 | 5.8 | 7.9 | 3.6 | – | – | – | – | – | – |
| GraphLE | 12 | 5.2 | 6.7 | 5.8 | 45% | 0.5 | 6.3 | 0.2 | 2.2 | **95**% |
| DiGress | 1.7±0.1 | 5.0±0.1 | **3.6**±0.4 | **1.6**±0.1 | 74%±4 | 0.8±0.0 | 4.1±0.3 | 0.5±0.0 | **1.2**±0.4 | 76%±1 |
| SparseDiff | 1.6±0.9 | 5.0±0.1 | 4.5±0.9 | 1.7±0.5 | 75%±10 | **0.3**±0.0 | **3.2**±0.3 | **0.1**±0.1 | **1.2**±0.4 | 85%±9 |

**Appendix C.2.** To further highlight our model's scalability, we include the Facebook dataset, which contains 1,045 nodes, in Table 11, and the CORA dataset, which contains 2,485 nodes, in Table 12, respectively.

**Metrics**  For evaluation, we use maximum mean discrepancy (MMD) metrics, standard in graph generation tasks. We report the validity of SBM graphs as the fraction passing a stochastic block model test, and for Planar graphs, the fraction that are planar and connected. For larger datasets, we also use the Radial Basis Function (RBF) MMD metric to assess fidelity and diversity using a randomly parametrized GNN (Thompson et al., 2022). Since MMD metrics often yield small values that are difficult to compare directly, we report Degree, Cluster, Orbit, Spectre and RBF MMD metrics in units of $10^{-3}$, $10^{-2}$, $10^{-2}$, $10^{-3}$ and $10^{-2}$, respectively. The theoretical optimal metrics, computed with $\text{MMD}(\text{train}, \text{test})^2$, are used as the reference and represented by a light gray line. Detailed results with higher precision are available in Appendix C.3 for facilitating comparison. For the Facebook and the CORA dataset, the evaluation follows the setup of SaGess (Limnios et al., 2023) and EDGE (Chen et al., 2023) for better comparison.

**Results**  Tables 1 and 2 show that SparseDiff tops the aggregated average-ratio metrics, trailing DiGress by just 0.1 on SBM. SparseDiff matches DiGress's best score within variance, suggesting it can hit that mark with an optimal seed. Dense models such as DiGress (Vignac et al., 2023a) excel on SBM and Planar graphs but falter on larger Ego and Protein datasets because of quadratic memory, failing outright for graphs over 1,000 nodes. SparseDiff, by tuning the sparsity parameter $\lambda$, allows larger batches under the same resources, converges faster, and handles graphs well past this threshold. On mid-sized SBM and Planar graphs it matches dense and sparse baselines (Table 2); it stays competitive on large Ego and Protein graphs (Table 1) and leads on the 1,000- and 2,000-node benchmarks (Tables 11, 12).

Table 3: Molecule generation on QM9 with implicit hydrogens.

| Dense Models | | | | Sparse Models | | | |
|---|---|---|---|---|---|---|---|
| Model | Valid (%)↑ | Unique (%)↑ | Conn. (%)↑ | FCD ↓ | Model | Valid (%)↑ | Unique (%)↑ | Conn. (%)↑ | FCD ↓ |
| SPECTRE | 87.3 | 35.7 | - | - | GraphARM | 90.3 | - | - | 1.22 |
| GraphNVP | 83.1 | 99.2 | - | - | EDGE | 99.1 | **100** | - | 0.46 |
| GDSS | 95.7 | 98.5 | - | 2.9 | HGGT | 99.2 | 95.7 | - | 0.40 |
| DiGress | **99.3**±0.0 | 95.9±0.2 | 99.4±0.2 | 0.15±0.01 | SparseDiff | 99.2±0.1 | 96.4±0.1 | **99.8**±0.1 | **0.12**±0.00 |

Table 4: Drug-sized molecular generation on the MOSES dataset.

| Model | Valid (%) ↑ | Unique (%) ↑ | Novel (%) ↑ | Filters (%) ↑ | FCD ↓ | SNN (%) ↑ | Scaf (%) ↑ | Frag (%) ↑ |
|---|---|---|---|---|---|---|---|---|
| GraphINVENT | **96.4** | 99.8 | – | 95.0 | 1.22 | **53.9** | 12.7 | 98.6 |
| DiGress | 85.7 | **100.0** | 95.0 | **97.1** | **1.19** | 52.2 | 14.8 | 99.6 |
| SparseDiff | 84.7±0.2 | **100.0**±0.0 | **95.1**±0.1 | 97.0±0.2 | 1.28±0.01 | 52.2±.00 | **15.5**±1.3 | **99.8**±0.0 |

## 4.2 Molecule Generation

**Dataset and metrics** Given that our method behaves like dense models in the limit case where $\lambda = 1$, it is expected to align with their performance on small graph datasets. We evaluate our approach using the QM9 and MOSES molecular datasets, anticipating its performance comparable to that of dense models. The QM9 dataset (Wu et al., 2018) features molecules with up to 9 heavy atoms, while the MOSES benchmark (Polykovskiy et al., 2020), derived from ZINC Clean Leads, includes drug-sized molecules with extensive assessment tools. In QM9, we add formal charges as discrete node features during diffusion, similar to Vignac et al. (2023b), and apply the same to DiGress for consistency. We assess molecular performance by the proportion of connected graphs, validity of the largest connected component verified by RDKit, and uniqueness of over 10,000 molecules. Additionally, we use the Frechet ChemNet Distance (FCD) (Preuer et al., 2018) to measure molecular similarity, excluding 0.96% of invalid molecules for FCD analysis.

**Results** Table 3 demonstrates that SparseDiff consistently outperforms other scalable methods on the FCD metric, highlighting its effectiveness for small, structured graphs even without significant sparsity advantages. Additionally, SparseDiff achieves results comparable to the state-of-the-art dense model, DiGress, across other metrics on the QM9 dataset. Furthermore, Table 4 shows that on the drug-sized molecular dataset MOSES, SparseDiff maintains similar performance to DiGress, further validating its effectiveness.

## 4.3 Efficiency Analysis

Table 5: Unconditional generation on QM9 under different sparsity parameters $\lambda$.

| $\lambda$ | Valid↑ | Unique↑ | Connected↑ | FCD↓ |
|---|---|---|---|---|
| 100% | 99.2±0.1 | 96.4±0.1 | 99.8±0.1 | 0.12±0.00 |
| 50% | 99.1±0.1 | 96.8±0.2 | 99.6±0.1 | 0.11±0.01 |
| 25% | 99.2±0.1 | 96.5±0.2 | 99.6±0.1 | 0.12±0.01 |
| 10% | 99.1±0.1 | 96.9±0.2 | 99.6±0.0 | 0.11±0.00 |

Figure 5: Occupied GPU under different sparsity parameter $\lambda$.

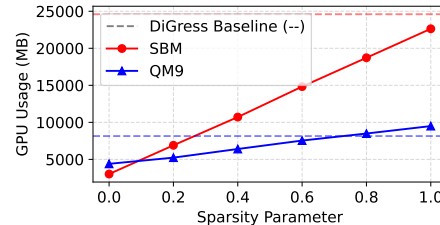

**Impact of the sparsity parameter** The sparsity parameter $\lambda$ plays a crucial role in SparseDiff, but its impact on performance is minimal. On the QM9 dataset, where $\lambda = 1.0$ (dense diffusion) is computationally feasible, we vary $\lambda$ from 1.0 to 0.1 (high sparsity) and report results in Table 5. SparseDiff demonstrates consistent performance across this range, with connectivity, validity, uniqueness, and Frechet ChemNet Distance (FCD) metrics showing minimal variation. For instance, the validity metric remains between 99.1% and 99.2%, while uniqueness ranges from 96.4% to 96.9%. We note that models trained with $\lambda = 0.1$ and $\lambda = 0.25$ required twice as many training epochs due to fewer edges being processed per epoch. Despite

this, all models exhibit consistent performance across different $\lambda$ values after convergence, which highlights the robustness and stability of SparseDiff in generating high-quality molecular graphs.

In Figure 5, we present the approximate linear relationship between the sparsity parameter $\lambda$ and the actual space complexity on the QM9 and SBM datasets, demonstrating effective control over space complexity through our proposed method. For example, on the SBM dataset, training with $\lambda = 0.25$ reduces memory usage to only 31.8% of that required by DiGress. At $\lambda = 1.0$, where sparsity is maximal, our method further outperforms DiGress by avoiding the overhead associated with dense models, which enforce a uniform size of $(n_{max}, n_{max})$ for batched computations, where $n_{\max}$ represents the largest node count within the batch. Conversely, on the QM9 dataset, where graph sizes are more uniform, our method incurs a slightly higher space complexity at $\lambda = 1.0$ due to the indexing operations inherent to message-passing mechanisms and the sparse graph representations.

Table 6: Convergence comparison of graph diffusion models after 24 hours of training.

| Model | Deg.↓ | Clust.↓ | Orbit↓ | $\overline{\text{Ratio}}$ ↓ | RBF↓ |
|---|---|---|---|---|---|
| EDP-GNN | 22 | 36 | 9.9 | 59 | - |
| DiGress | 4.0 | 4.9 | **3.4** | 11 | 5.6 |
| EDGE | 46 | 18 | 4.5 | 87 | 3.6 |
| GraphLE | 58 | 23 | 4.2 | 110 | - |
| SparseDiff | **2.3** | **4.7** | 3.6 | **7.8** | **3.5** |

Figure 6: Convergence comparison between DiGress and SparseDiff.

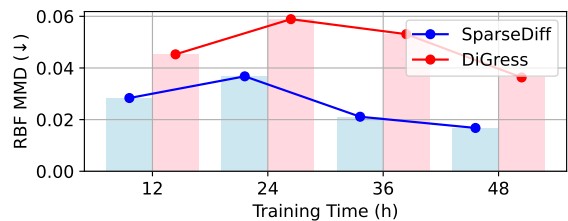

**Training efficiency** We compare SparseDiff's performance on the Ego dataset against other diffusion models, including dense models like EDP-GNN and DiGress, and sparse models such as EDGE and GraphLE. Table 6 shows that after 24 hours of training on a V100-32G machine, SparseDiff outperforms all metrics except for a minor increase in Orbit MMD compared to DiGress. The comparison of graph diffusion models' sampling speeds is further presented in Table 15 in Appendix D.4. Figure 6 shows that SparseDiff has a significantly faster convergence speed compared to DiGress, achieving satisfactory results within two days. Notably, a SparseDiff model trained for 12 hours demonstrates an RBF MMD comparable to a DiGress model trained for 48 hours.

**Inference efficiency** We test the Ego and Planar datasets with different numbers of inference steps (1000, 500, and 200) after training with $T = 1,000$ steps. The results, presented in Table 7, surprisingly demonstrate that even with a 5-fold increase in generation speed (down to 200 steps), our model keeps performing, and consistently outperforms most other dense and scalable models, highlighting its potential for more efficient generation.

Table 7: Inference acceleration results.

| Dataset | Steps | Deg.↓ | Clust.↓ | Orbit ↓ | $\overline{\text{Ratio}}$ ↓ | RBF↓ |
|---|---|---|---|---|---|---|
| Ego | 1000 | 3.6 | 3.1 | 1.5 | 8.2 | 2.0 |
| | 500 | 2.3 | 2.9 | 2.0 | 6.2 | 2.1 |
| | 200 | 3.7 | 3.1 | 1.6 | 8.4 | 2.3 |

| Dataset | Steps | Deg. | Clust. | Orbit | $\overline{\text{Ratio}}$ | V.U.N |
|---|---|---|---|---|---|---|
| Planar | 1000 | 0.3 | 3.2 | 0.1 | 1.2 | 85% |
| | 500 | 0.3 | 3.4 | 0.2 | 1.5 | 80% |
| | 200 | 0.5 | 3.7 | 0.4 | 2.6 | 69% |

## 5 Conclusion

In this work, we introduced SparseDiff, a scalable discrete denoising diffusion model for graph generation. SparseDiff offers precise control over computational resources by predicting only a subset of edges at each step. Experimental results highlight its superior and robust performance across graphs of varying sizes, making it applicable to tasks such as generating large molecules and community graphs. Additionally, the query edge design, sparse transformer architecture, and iterative sampling procedure of SparseDiff can be extended to other iterative graph generation models. Furthermore, the efficient sampling algorithm for non-existing edges is applicable for graph rewiring in other domains. While SparseDiff meets the demands of most scenarios, its scalability and ability to generate graphs with out-of-distribution node counts could be further enhanced by incorporating a structured hierarchical approach, which is expected for future work.

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

# A  Theoretical Analysis

The notations used throughout the paper are summarized in Tab. 8.

| Notation | Explanation |
|---|---|
| $G = (\boldsymbol{E}, \boldsymbol{X}, \boldsymbol{Y})$ | Graph as (edge list, node attributes, edge attributes) |
| $n$ | Number of nodes |
| $m$ | Number of edges |
| $\boldsymbol{E} \in \mathbb{N}^{2 \times m}$ | Edge list (pairs of node indices) |
| $\boldsymbol{E}_{ne} \in \mathbb{N}^{2 \times (\binom{n}{2} - m)}$ | List of non-existing node-pairs |
| $\boldsymbol{X} \in \{0,1\}^{n \times a}$ | One-hot node attribute matrix |
| $\boldsymbol{Y} \in \{0,1\}^{m \times b}$ | One-hot existing-edge attribute matrix |
| $a, b$ | # classes for nodes ($a$) and edges ($b$) |
| $G^t$ | Noisy graph at diffusion step $t$ |
| $q(G^t \mid G^{t-1})$ | Markov transition probability from $G^{t-1}$ to $G^t$ |
| $\boldsymbol{Q}_{\boldsymbol{X}}^t, \boldsymbol{Q}_{\boldsymbol{Y}}^t$ | Transition matrices for node/edge classes at $t$ |
| $\alpha^t, \beta^t$ | $\alpha^t = 1 - \beta^t$, noise scheduling scalars |
| $\bar{\alpha}^t$ | $\prod_{s=1}^t \alpha^s$, cumulative noise schedule |
| $\bar{\boldsymbol{Q}}_{\boldsymbol{X}}^t, \bar{\boldsymbol{Q}}_{\boldsymbol{Y}}^t$ | Cumulative transition matrices |
| $\boldsymbol{p_X}, \boldsymbol{p_E}$ | Marginal class probabilities (nodes/edges) |
| $\bar{m}_t = \binom{n}{2} - m_t$ | # non-existing edges in $G^t$ |
| $q_t = 1 - \boldsymbol{Q}^t[0,0]$ | Probability for non-edge becoming an actual edge at step $t$ |
| $\boldsymbol{E}_q$ | Query edge set |
| $\lambda$ | Sparsity param. ($\lvert \boldsymbol{E}_q \rvert / n^2$) |
| $C_k$ | $k$-th chunk of node-pair queries |
| $\boldsymbol{E}_m = \boldsymbol{E}^t \cup \boldsymbol{E}_q$ | Message-passing edge list for denoising |
| $\phi_\theta$ | Sparse denoising network $\phi$ with parameters $\theta$ |
| $\hat{\boldsymbol{P}}_q^G = (\hat{\boldsymbol{P}}^X, \hat{\boldsymbol{P}}_q^Y)$ | Predicted node and query-edge distributions |
| $\hat{\boldsymbol{X}}, \hat{\boldsymbol{Y}}_q$ | Sampled labels via Multinomial($\hat{\boldsymbol{P}}^X$), Multinomial($\hat{\boldsymbol{P}}_q^Y$) |

Table 8: Main notations used in SparseDiff.

## A.1  Space Complexity Analysis

In this section, we provide a comprehensive breakdown of the space complexity of SparseDiff for each component, supporting our claim that SparseDiff achieves higher efficiency.

We consider a batch of graphs where each graph $i$ contains $n_i$ nodes and $m_i$ edges. Let $bs$ denote the batch size. To analyze worst-case complexity, we define:

$$n_{\max} = \max_{0 \leq i < bs} n_i, \quad m_{\max} = \max_{0 \leq i < bs} m_i. \tag{4}$$

### A.1.1  Efficient Noise Model

Our method explicitly exploits redundancy in the previous computation of $Y\bar{Q}^t \in \mathbb{R}^{n \times n \times b}$. Specifically, for edges sharing the same type (e.g., non-existing), the resulting noisy probabilities are identical. We leverage this property to eliminate repeated computations and substantially improve efficiency.

**Existing edges**  For existing edges, computing the noisy distribution involves matrix multiplication with a cost of $\mathcal{O}(2mb + b^2)$. Multinomial sampling from this distribution requires further $\mathcal{O}(mb)$. Therefore, the total complexity is $\mathcal{O}(3mb + b^2)$.

**Non-existing edges**  For non-existing edges, the sparse computation consists of three main steps. Based on the *PyTorch* implementation, the space complexity of each step is as follows:

1. **Sampling edge count**: $\mathcal{O}(1)$.

2. **Sampling positions**: $\mathcal{O}(2n^2 + m)$.

3. **Sampling edge attributes**: $\mathcal{O}(bm)$, where $b$ is the number of edge types.

The breakdown of space complexity for sampling positions is:

- **Steps 1 and 2**: A multinomial distribution is used to sample $\bar{m}_t$ positions from the $O(n^2 - m_t)$ non-existing edge candidates in $E_{ne}$. This results in space complexity $\mathcal{O}(n^2 - m_t + \bar{m}_t)$. Assuming $\bar{m}_t \approx m$ and $m_t \approx m$, this simplifies to $\mathcal{O}(n^2)$.

- **Step 3**: Offsets are appended to the candidate edge list using a single pass over the sampled edges. This step has space complexity $\mathcal{O}(n^2 - m_t)$. The space complexity to take into account existing edges is $\mathcal{O}(m_t)$. This results in space complexity $\mathcal{O}(n^2)$.

- **Step 4**: Final operation to convert these positions back to index pairs selection involves element-wise operations over $\bar{m}_t \sim m$ elements. These operations have $\mathcal{O}(m)$ memory overhead.

In total, the space complexity for handling non-existing edges is $\mathcal{O}(2n^2 + bm + m + 1)$. Steps 1 and 2 are detached from the backward computation graph and involve trivial operations that are typically well-supported by existing frameworks.

**Summary**    By aggregating all edges and adopting the same computation for existing edges as in SparseDiff, DiGress has a total space complexity of $\mathcal{O}(3n^2b + b^2)$. In contrast, SparseDiff requires $\mathcal{O}(1 + m + 4mb + b^2 + 2n^2)$. Given that $n$ is typically large and that $m$ is negligeble compared to $n^2$, we compare the highest-order terms. Specifically, DiGress has leading complexity $\mathcal{O}(3n^2b)$, while SparseDiff has $\mathcal{O}(2n^2)$.

Assuming $m$ is negligible relative to $n$ and taking $b = 2$ as a concrete example, the leading space complexity of DiGress becomes $\mathcal{O}(3n^2b) = \mathcal{O}(6n^2)$. In contrast, our optimized implementation eliminates redundant computations across edge types and achieves a total space complexity of $\mathcal{O}(2n^2)$. This results in a $3\times$ reduction in memory usage. As $b$ increases when we consider more edge types other than edge existance, the gap widens further, making our approach increasingly advantageous in such settings.

### A.1.2    Efficient Denoising Neural Network

As described in Section 3.2, the space complexity of the denoising neural network for each graph is determined by the number of edges used in its message-passing process:

$$|\boldsymbol{E}_{m,i}| \leq m_i + \lceil \lambda n_i^2 \rceil, \tag{5}$$

where $\lambda$ is the tunable sparsity parameter controlling the number of sampled query edges. For a batch, the total number of edges being considered is dominated by $bs|\boldsymbol{E}_{m,\max}|$, where $|\boldsymbol{E}_{m,\max}| = m_{\max} + \lceil \lambda n_{\max}^2 \rceil$:

$$\sum_{0 \leq i < bs} |\boldsymbol{E}_{m,i}| = \sum_{0 \leq i < bs} m_i + \lceil \lambda n_i^2 \rceil \leq bs(m_{\max} + \lceil \lambda n_{\max}^2 \rceil) = bs|\boldsymbol{E}_{m,\max}|. \tag{6}$$

Traditional full-attention mechanisms considers all node pairs which incurs as a complexity linear to $\mathcal{O}(bs \cdot n_{\max}^2)$, whereas SparseDiff scales only with a complexity upper-bounded by $\mathcal{O}(bs \cdot (m_{\max} + \lambda n_{\max}^2))$.

In practice, for a network with $L$ attention layers and edge activation dimension $d_e$, the total space complexity for a batch of size $bs$ is actually upper-bouned by:

$$\mathcal{O}(bs \cdot L \cdot d_e \cdot |\boldsymbol{E}_{m,\max}|). \tag{7}$$

Since $d_e$ is typically large (e.g., 64 or 128) and $L$ often exceeds five layers, this component becomes the primary memory bottleneck inside the network. In sparse graphs, the number of edges $m_i$ is typically much

smaller than $n_i^2$, making $\lambda n_{\max}^2$ the dominant term in $|\boldsymbol{E}_{m,\max}|$. As a result, the space complexity grows as $\mathcal{O}(\lambda n_{\max}^2)$ in practice, leading to a near-linear scaling of memory usage with $\lambda$. This is further supported by the observation in Figure 5 that the memory usage scales approximately linearly with $\lambda$, as this component dominates the GPU usage.

### A.1.3 Iterative Inference

SparseDiff employs an iterative inference process that aligns with its training procedure, to maintain sparsity in message-passing edges. This contrasts with previous models that rely on dense attention maps constructed from all node pairs. The complexity thus remains upper-bounded by $\mathcal{O}(bs \cdot L \cdot d_e \cdot |\boldsymbol{E}_{m,\max}|)$.

### A.1.4 Summary

We provide a summary of the highest-order complexity for the three components of SparseDiff:

- **Efficient Noise Model:** SparseDiff reduces noise application complexity to $\mathcal{O}(2n^2)$ instead of $\mathcal{O}(3bn^2)$, $b \geq 2$ by avoiding redundant computations and leveraging sparsity in non-existing edge processing.

- **Efficient Denoising Neural Network:** SparseDiff's denoising network operates over a sparse attention map, scaling as $\mathcal{O}(\lambda n^2)$, unlike dense baselines (e.g., DiGress), which process a full attention map with $\mathcal{O}(n^2)$ complexity.

- **Iterative Inference:** Inference reuses sparsely sampled edges from training, keeping its space complexity similarly scaling with $O(\lambda n^2)$.

Among them, the denoising network is the primary memory bottleneck during training. In contrast to dense baselines such as DiGress, which scale as $O(n^2)$ with a large constant $d_e \cdot L$ with $L$ as the number of layers and $d_e$ as the dimension of edge activations, typically set to 64 or 128, SparseDiff constrains memory usage by operating on a sparse set of sampled edges scaling with $O(\lambda n^2)$. This is further supported by Fig.5 in the revised manuscript, which shows that GPU memory usage scales approximately linearly with $\lambda$.

### A.2 Proof of Lemma 3.1

The lemma for a noisy graph with guaranteed sparsity comes directly from the proposition regarding the tail behavior of a binomial distribution (Desolneux et al., 2008) as follows:

**Proposition A.1.** *(Tail behavior of a binomial distribution)*

*Let $X_i, i = 1, ...l$ be independent Bernoulli random variables with parameter $0 < p < \frac{1}{4}$ and let $S_l = \sum_{i=1}^{l} X_i$. Consider a constant $p < r < 1$ or a real function $p < r(l) < 1$. Then according to the Hoeffding inequality, $\mathcal{B}(l, k, p) = \mathbb{P}[S_l \geq k]$ satisfies:*

$$-\frac{1}{l} log\mathcal{P}[S_l \geq rl] \geq r log \frac{r}{p} + (1-p) log \frac{1-r}{1-p} \tag{8}$$

For sparse graphs, the edge ratio $r$ is clearly smaller than $\frac{1}{4}$. Consider then Bernoulli random variables with the parameter $r$ and a noised edge ratio $r < k < 1$ with $l = n(n-1)/2$ (i.e. number of all node pairs in an undirected graph without self-loops) draws, and note the sampled 'existing' edge number $S_{n(n-1)/2}$ as $m_t$, we have:

$$log(\mathbb{P}[r_t = \frac{m_t}{n(n-1)/2} \geq k]) \leq -\frac{n(n-1)}{2}[k \log \frac{k}{r} + (1-r) \log \frac{1-k}{1-r}] \tag{9}$$

# B   Model Architecture

Our sparse denoising network adopts the graph transformer architecture Veličković et al. (2017), featuring normalization, feed-forward, and attention layers. To handle sparse data, it incorporates the sparse attention mechanism (Shi et al., 2020) based on weighted message-passing layers and integrates enhancements from Vignac et al. (2023a), such as PNA pooling layers (Corso et al., 2020) and FiLM layers (Perez et al., 2018).

Precisely, we introduce the FiLM layer and the PNA layer inside the model architecture to enhance its performance. Precisely, the FiLM layer is used to combine features at different scales, such as node and edge features. Specifically, given two features $M_1$ and $M_2$, and trainable parameters $W_1$ and $W_2$, the FiLM layer output is calculated as $\text{FiLM}(M_1, M_2) = M_1 W_1 + (M_1 W_2) \odot M_2 + M_2$. As an illustration, within the convolutional layer, the graph feature $M_2$ is integrated with edge features $M_1$ to enhance predictions. While PNA layer is used as a specialized pooling layer to obtain information from different dimensions of a specific feature. Given the feature X and trainable parameter $W$, $\text{PNA}(X) = \text{cat}(\max(X), \min(X), \text{mean}(X), \text{std}(X)) \, W$. For example, node features $X$ are forwarded to a PNA layer for extracting global information across different scales, which is subsequently added to the graph feature to enhance its representation.

Finally, we enrich the message-passing graph with additional encodings, such as the graph Laplacian and cycle counts, to enhance structural and positional information (detailed in Appendix B.1). These encodings can only be computed effectively when the noisy graphs are sparse, which is another significant advantage of discrete diffusion models using marginal transitions. It is worth noting that not all these encodings can be computed in sub-quadratic time. However, in practice, this does not pose an issue as they are not used for back-propagation, which arises as the primary complexity bottleneck during training. For instance, for the large Protein dataset, computing these encodings is five times faster than the forward pass itself. Nevertheless, on very large graphs, these expensive encodings should not be computed.

## B.1   Additional Encodings

During training, we augment model expressiveness with additional encodings. To make things clear, we divide them into encodings for edges, nodes, and for graphs.

**Encoding for graphs**   We first incorporate graph eigenvalues, known for their critical structural insights, and cycle counts, addressing message-passing neural networks' inability to detect cycles (Chen et al., 2020). The first requires $n^3$ operations for matrix decomposition, the second requires $n^2$ for matrix multiplication, but both are optional in our model and do not significantly limit scalability even with graphs up to size 500. In addition to the previously mentioned structural encodings, we integrate the degree distribution to enhance the positional information within the graph input, which is particularly advantageous for graphs with central nodes or multiple communities. Furthermore, for graphs featuring attributed nodes and edges, the inclusion of node type and edge type distributions also provides valuable benefits.

**Encoding for nodes**   At the node level, we use graph eigenvectors, which are fundamental in graph theory, offering valuable insights into centrality, connectivity, and diverse graph properties.

**Encoding for edges**   To aid in edge label prediction, we introduce auxiliary structural encodings related to edges. These include the shortest distance between nodes and the Adamic-Adar index. The former enhances node interactions, while the latter focuses on local neighborhood information. Due to computational constraints, we consider information within a 10-hop radius, categorizing it as local positional information.

**Molecular information**   In molecular datasets, we augment node features by incorporating edge valency and atom weights. Additionally, formal charge information is included as an additional node label for diffusion and denoising during training, as formal charges have been experimentally validated as valuable information (Vignac et al., 2023b).

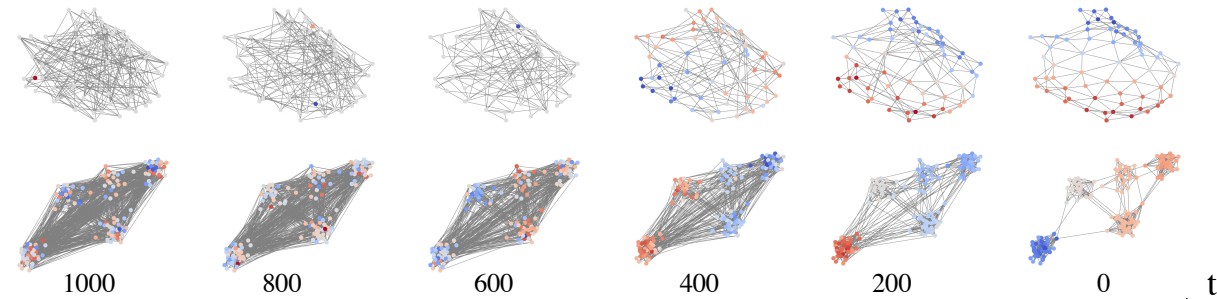

Figure 7: Visualization of iterative generation for Planar and SBM graphs.

## C    Experimental setup

In our experimental setup, we utilize a single V100-32G GPU machine, which is particularly prone to scalability issues, to demonstrate that our method allows users with limited GPU resources to effectively train on larger graphs. Detailed specifications regarding workers, memory allocation, execution time, and optimizers are meticulously indicated in the configuration details provided in our code.

As for dataset splits, we adhere to the framework established by DiGress. Specifically, for the QM9 dataset, we implement a split comprising 100k molecules for training, 20k for validation, and 13k for evaluating likelihood on the test set. For the Planar, SBM, and Protein datasets, employing a seed of 1234, we randomly assign 20% of the graphs to testing, while 80% of the remaining graphs are utilized for training, and 20% for validation. For the Ego dataset, to ensure consistency with previous methods and a fair comparison, we maintain a split of 80% for training and 20% for testing, with 20% of the training set additionally used for validation purposes. All configuration details are comprehensively documented in the code provided.

### C.1    MMD metrics

In our research, we carefully select specific metrics tailored to each dataset, with a primary focus on four widely recognized Maximum Mean Discrepancy (MMD) metrics. These metrics use the total variation (TV) distance, as detailed in Martinkus et al. (2022). They encompass node degree (Deg), clustering coefficient (Clus), orbit count (Orb), and graph spectra (Spec). The first three local metrics compare the degree distributions, clustering coefficient distributions, and the occurrence of all 4-node orbits within graphs between the generated and training samples. Additionally, the comparison of graph spectra is realized by computing the eigenvalues of the normalized graph Laplacian, providing complementary insights into the global properties of the graphs.

### C.2    Statistics of the datasets

Table 9: Statistics for the datasets employed in our experiments.

| Name | Graph number | Node range | Edge range | Edge Ratio (%) | $\lambda$ (%) |
|---|---|---|---|---|---|
| QM9 | 133,885 | [2,9] | [2, 28] | [20, 56] | 100 |
| QM9(H) | 133,885 | [3, 29] | [4, 56] | [7.7, 44] | 50 |
| MOSES | 1,936,962 | [8, 27] | [14, 62] | [8.0, 22] | 50 |
| Planar | 200 | [64, 64] | [346, 362] | [8.4, 8.8] | 50 |
| SBM | 200 | [44, 187] | [258, 2258] | [6.0, 17] | 25 |
| Ego | 757 | [50, 399] | [112, 2124] | [1.2, 11] | 10 |
| Protein | 918 | [100, 500] | [372, 3150] | [8.9, 6.7] | 10 |

To provide a more comprehensive overview of the various scales found in 'existing' graph datasets, we present here key statistics for them. These statistics encompass the number of graphs, the range of node numbers, the range of edge numbers, the edge ratio for 'existing' edges, and the sparsity parameter $\lambda$ used for training, i.e. the proportion of 'existing' edges among all node pairs. In our consideration, we focus on undirected graphs. Therefore, when counting edges between nodes $i$ and $j$, we include the edge in both directions.

### C.3 Raw results

Table 10: Raw results on the SBM, Planar, Protein, and Ego datasets.

| Model | Deg. (e-3)↓ | Clust. (e-2)↓ | Orbit (e-2)↓ | Spec. (e-3)↓ | FID↓ | RBF MMD (e-2)↓ |
|---|---|---|---|---|---|---|
| *SBM* | | | | | | |
| Training set | 0.85 | 3.32 | 2.55 | 2.74 | 1.37 | 3.23 |
| SparseDiff | 1.57±0.91 | 5.04±0.06 | 4.51±0.90 | 6.68±2.04 | 4.55±2.01 | 4.98±0.06 |
| *Planar* | | | | | | |
| Training set | 0.19 | 3.10 | 0.05 | 3.82 | 1.57 | 8.89 |
| SparseDiff | 0.32±0.01 | 3.25±0.35 | 0.09±0.08 | 6.99±0.92 | 2.94±3.15 | 9.84±0.91 |
| *Protein* | | | | | | |
| Training set | 0.32 | 0.68 | 0.32 | 0.49 | 1.36 | 1.37 |
| SparseDiff | 1.45±0.30 | 3.35±0.33 | 0.53±0.78 | 1.35±0.16 | 5.97±1.07 | 3.77±0.65 |
| *Ego* | | | | | | |
| Training set | 0.16 | 0.71 | 0.69 | 0.98 | 0.07 | 0.86 |
| SparseDiff | 3.70±0.44 | 3.18±0.10 | 1.98±0.42 | 5.63±0.80 | 4.84±1.56 | 2.60±0.31 |

To ease comparison with other methods, Table 10 provides the raw numbers (not presented as ratios) for the SBM, Planar, Ego, and Protein datasets. Note that this table contains the FID metrics from Thompson et al. (2022), which we did not include in the main text. The reason is that we found this metric to be very brittle, with some evaluations giving a very large value that would dominate the mean results. Besides, we have identified a discrepancy in the Spectre metrics reported in the study by Martinkus et al. (2022) when computed under non-parallel computation. We thus provide the updated values for reference and use the updated value for ratio calculation in Table 2 and in Table 1.

## D   Additional experiments

### D.1   Training with larger graphs

However, using the same graph for both training and evaluation poses potential risks, as high performance metrics could merely reflect overfitting to the training graph. Therefore, we report results on only the two relevant datasets in the appendix.

Table 11: Large graph generation on the Facebook dataset. Triangles and squares are abbreviated as 'tri' and 'squ' in the table, while PLE represents power law exp.

| Model | Num Nodes | Num Edges | Num Triangles | Num Squares | Max Deg | Clust | Assort | PLE | CPL |
|---|---|---|---|---|---|---|---|---|---|
| Ref | 1045 | 27,755 | 446,846 | 34,098,662 | 1044 | 0.57579 | -0.02543 | 1.28698 | 1.94911 |
| SAGESS-Uni | 1043 | 27,758 | 429,428 | 35,261,545 | 999 | 0.52098 | -0.01607 | 1.29003 | 2.00800 |
| SAGESS-RW | 1009 | 27,764 | 490,844 | 43,006,252 | 1001 | **0.56138** | -0.02266 | 1.29398 | 1.96014 |
| SAGESS-Ego | 1005 | 27,761 | 515,928 | 45,421,130 | 295 | 0.43074 | 0.34074 | 1.29381 | 2.65926 |
| NetGAN | **1045** | **27,755** | 262,574 | 15,635,626 | 849 | 0.39773 | -0.01821 | 1.27429 | 2.13730 |
| CELL | **1045** | **27,755** | 250,968 | 14,855,676 | 474 | 0.30854 | 0.12788 | 1.27490 | 2.38650 |
| DCSBM | 1041 | 27,092 | 339,448 | 26,714,948 | 733 | 0.37549 | 0.07125 | 1.28845 | 2.33021 |
| SaGess | 1043 | 27,758 | 429,428 | 35,261,545 | 999 | 0.52098 | -0.01607 | 1.29003 | 2.00800 |
| SparseDiff | **1045** | 27,763 | **446,819** | **34,095,513** | **1044** | 0.43310 | **-0.02536** | **1.28687** | **1.94921** |

We first train on the large graph with 1045 nodes from the Facebook dataset, following the SaGess (Limnios et al., 2023) setting. SparseDiff was evaluated using SaGess metrics as a reference. In the provided table, we present SaGess-RW, demonstrating the best results among the three proposed SaGess models. Notably, SaGess generates small graphs and concatenates them to meet the required number of edges, while SparseDiff generates a single large graph based on the specified node count. This explains SparseDiff's advantage in the 'num nodes' metric and SaGess's advantage in the 'num edges' metric. Furthermore, SparseDiff closely aligns with real data statistics, except for the clustering coefficient, showcasing not only its scalability up to 1000 nodes but also its strong performance on such single-graph datasets.

Table 12: Large graph generation on the CORA dataset.

| Model | EO | PLE | NTC | CC | CPL | AC |
|---|---|---|---|---|---|---|
| Ref | 100 | 1.885 | 1 | 0.090 | 6.311 | -0.071 |
| OPB | 10.9 | 1.852 | 0.097 | 0.008 | 4.476 | -0.037 |
| HDOP | 0.9 | 1.849 | 0.113 | 0.009 | 4.770 | -0.030 |
| CELL | 10.3 | 1.774 | 0.009 | 0.002 | 5.799 | -0.018 |
| CO | 9.7 | 1.776 | 0.009 | 0.002 | 5.653 | **0.010** |
| TSVD | 6.7 | 1.858 | 0.349 | **0.082** | 4.908 | -0.006 |
| VGAE | 1.5 | 1.717 | 0.120 | 0.220 | 4.934 | 0.002 |
| GRNN | **0.4** | 1.822 | 0.043 | 0.011 | **6.146** | 0.043 |
| EDGE | 1.1 | 1.755 | 0.446 | 0.034 | 4.995 | **-0.046** |
| SparseDiff (pos) | 0.3 | **1.896** | **1.434** | 0.075 | 4.747 | -0.043 |

We evaluate our model on the CORA dataset (McCallum et al., 2000), as used in EDGE (Chen et al., 2023). The CORA graph consists of 2,485 nodes. Our model is trained with positional encoding for a fair comparison of the edge overlap ratio (EO), with the sparsity parameter set to 0.05. The performance results, after just one day of training, are presented in the table below. As shown, SparseDiff outperforms EDGE in 3 out of 5 metrics and remains competitive in the AC metric, further validating the scalability of our method.

### D.2 QM9 with explicit hydrogens

Table 13: Unconditional generation on QM9 with explicit hydrogens. On small graphs such as QM9, sparse models are not beneficial, but SparseDiff still achieves very good performance.

| Model | Connected | Valid↑ | Unique↑ | Atom stable↑ | Mol stable↑ |
|---|---|---|---|---|---|
| DiGress | – | 95.4 | **97.6** | 98.1 | 79.8 |
| DiGress + charges | **98.6** | 97.7 | 96.9 | 99.8 | 97.0 |
| SparseDiff(ours) | 98.3$\pm$.08 | **97.9**$\pm$.13 | 97.4$\pm$.10 | - | - |

We additionally report the results for QM9 with explicit hydrogens in Table 13. Having explicit hydrogens makes the problem more complex because the resulting graphs are larger. We observe that SparseDiff achieves better validity than DiGress and has comparable results on other metrics when both are utilizing charges.

### D.3 MOSES benchmark evaluation

Table 14: Mean and standard deviation across 5 samplings on the MOSES benchmark. SparseDiff has a similar performance to DiGress, despite a shorter training time.

| Model | Connected ↑ | Valid (%) ↑ | Unique (%) ↑ | Novel (%) ↑ | Filters (%) ↑ |
|---|---|---|---|---|---|
| GraphINVENT | – | **96.4** | 99.8 | – | 95.0 |
| DiGress | – | 85.7 | **100.0** | 95.0 | **97.1** |
| SparseDiff | 94.8±.1 | 84.7±.2 | **100.0**±.0 | **95.1**±.1 | **97.0**±.2 |

| Model | FCD ↓ | SNN (%) ↑ | Scaf (%) ↑ | Frag (%) ↑ | IntDiv (%) ↑ |
|---|---|---|---|---|---|
| GraphINVENT | 1.22 | **53.9** | 12.7 | 98.6 | **85.7** |
| DiGress | **1.19** | 52.2 | **14.8** | 99.6 | 85.3 |
| SparseDiff | 1.28±.01 | 52.2±.0 | **15.5**±1.3 | **99.8**±.0 | 85.4±.0 |

| Model | IntDiv2 (%) ↑ | logP $(e^{-2})$ ↓ | SA$(e^{-2})$ ↓ | QED $(e^{-3})$ ↓ | Weight (%) ↓ |
|---|---|---|---|---|---|
| GraphINVENT | **85.1** | **0.67** | 4.5 | **0.25** | 16.1 |
| DiGress | – | 3.4 | **3.6** | 2.91 | **1.42** |
| SparseDiff | 84.8±.0 | 3.0±.3 | 5.4±.2 | 1.21±.21 | 5.58±.15 |

MOSES is an extensive molecular dataset with larger molecular graphs than QM9, offering a much more comprehensive set of metrics. While autoregressive models such as GraphINVENT are recognized for achieving higher validity on this dataset, both SparseDiff and DiGress exhibit advantages across most other metrics. Notably, SparseDiff closely aligns with the results achieved by DiGress, affirming the robustness of our method on complex datasets.

### D.4 Sampling Speed Comparison

Table 15: Sampling speed for generating 8 Ego graphs.

| Model | EDP-GNN | DiGress | EDGE | GraphLE | SparseDiff | SparseDiff (200 steps) |
|---|---|---|---|---|---|---|
| Time (min) | 5 | 32 | 2 | 20 | 28 | 5 |

The speed of generating 8 Ego graphs is demonstrated in Table 15. Notably, for GraphLE, the batch size is constrained to 2 due to its substantial memory requirements. An additional column labeled "SparseDiff (200 steps)" represents the sampling time after reducing the inference steps from 1,000 to 200 through acceleration strategies. The table illustrates that SparseDiff maintains comparable speed to dense models without significant compromise on space efficiency and can be significantly accelerated during sampling.

### D.5 Ablations

This part presents 2 ablation experiments that motivate our approach. SparseDiff builds upon an experimental observation and a hypothesis. Firstly, our experiments demonstrate that relying solely on node features for link prediction yields unsatisfactory results. This observation encouraged us to design the message-passing graph that contains all edges to be predicted (i.e. query edges) as the message-passing graph to directly obtain their edge features. Secondly, we hypothesized that preserving the same distribution of edge types as observed in dense graphs for loss calculation is advantageous for training. This hypothesis necessitates the sampling of query edges within each graph in a batch of graphs with varying sizes, thereby introducing increased complexity to the algorithm design process.

### D.5.1 Link Prediction

Table 16: Influence of including edges features for edge prediction.

| Model | Deg ↓ | Clus ↓ | Orb↓ | Spec↓ | FID↓ | RBF MMD↓ |
|---|---|---|---|---|---|---|
| Link Pred | 0.0043 | 0.0721 | **0.0275** | 0.0344 | 1.51e6 | **0.0315** |
| SparseDiff | **0.0019**±.00 | **0.0537**±.00 | 0.0299±.00 | **0.0050**±.00 | **16.1**±12.9 | 0.0483±.01 |

Table 17: Influence of including edges features for edge prediction on the QM9 dataset.

| Model | Valid↑ | Unique↑ | Connected↑ | FCD↓ |
|---|---|---|---|---|
| Link Pred | 98.12 | 96.25 | 99.58 | 0.310 |
| SparseDiff | **99.23**±0.06 | **96.37**±0.13 | **99.76**±0.06 | **0.117**±0.004 |

In Table 16, the model that does not explicitly incorporate edge features for edge prediction underperforms across all metrics, except for RBF MMD and orbit. Similarly, in Table 17, the link prediction-based method fails to achieve comparable validity, even though QM9 is widely recognized as an easy dataset to learn. Both experiments highlight a subtle yet challenging gap between message-passing and transformer-based architectures for graph generation, as the latter provides richer topological interactions. While, in our case, this ablation proves detrimental, developing a more robust link prediction module could simplify the task to link prediction and significantly reduce space complexity.

### D.5.2 Query edges with proper distribution

Table 18: Influence of edge loss distribution on EGO dataset.

| Loss based on | Deg ↓ | Clus ↓ | Orb↓ | Spec↓ | FID↓ | RBF MMD↓ |
|---|---|---|---|---|---|---|
| Comp graph | 0.0021 | 0.0566 | **0.0270** | 0.0100 | 28.2 | **0.0396** |
| Query graph | **0.0019**±.00 | **0.0537**±.00 | 0.0299±.00 | **0.0050**±.00 | **16.1**±12.9 | 0.0483±.01 |

In order to emphasize the importance of preserving the edge distribution when computing losses, we conduct an experiment where we assess the performance of a model trained using all message-passing edges as opposed to solely using query edges. The former results in an increased emphasis on 'existing' edges during training compared to SparseDiff. Similarly, we use the Ego dataset for initial experiments. Table 18 shows that using edges of the message-passing graph $G_m$ results in worse performance on most of the metrics, which indicates the importance of keeping a balanced edge distribution for loss calculation.

### D.6 Hyperparameters

**Training setup** The model is trained using a diffusion-based framework consisting of 1,000 denoising steps. To balance the objectives associated with node and edge predictions, fixed loss coefficients of 5 and 2 are applied to edge and node terms, respectively. The learning rate is set to a constant value of 0.0002 across all datasets. All experiments are conducted on a single V100 GPU with 32GB memory. The only introduced hyperparameter is the sparsity controller $\lambda$, which is selected based on graph size according to Tab. 5.

**Model architecture** For large graph datasets, the model comprises 8 sparse convolutional attention layers, following the same setup as the dense attention version proposed by Vignac et al. (2023a). It employs a multi-head attention mechanism with 8 heads to jointly process 256-dimensional node features, 64-dimensional edge features, and 128-dimensional graph-level features. Hidden dimensions for the MLP layers before and after the transformer block are set to 256 for nodes, 64 for edges, and 128 for graphs. For molecular

and synthetic datasets, we adopt the same architectural configuration as DiGress (Vignac et al., 2023a) to maintain consistency across experimental settings.

Additional hyperparameters are further available in the shared codebase.

# E   Comparison to Baseline Methods

## E.1   Scalable Methods

**EDGE**   EDGE (Chen et al., 2023) uses absorbing states to construct sparse diffusion by first generating a node degree distribution $\boldsymbol{d}^0$ and then building the adjacency matrix $\boldsymbol{A}$ based on node degree changes during inference. While this factorization is broadly applicable, the conditional distribution $p_\theta(\boldsymbol{A}|\boldsymbol{d}^0)$ can yield degree sequences that are invalid for undirected graphs, creating a mismatch between training and sampling. EDGE also underperforms compared to SparseDiff on both small and large graphs. In contrast, SparseDiff only assumes the sparsity of graphs, ensuring alignment between training and generation.

**DruM**   DruM (Jo et al., 2023) predicts a destination graph and mixes several diffusion processes conditioned on that structure, introducing strong topological priors that guide denoising and improve convergence. However, DruM operates within a continuous diffusion framework. SparseDiff, by contrast, relies entirely on discrete diffusion, which better aligns with the inherently discrete nature of graph data.

**BiGG**   BiGG (Dai et al., 2020) generates graphs autoregressively by hierarchically decomposing the node-pair space into a binary tree, achieving $O((n+m)\log n)$ sampling complexity. This structure enforces a fixed generation order. SparseDiff avoids such inductive constraints, reconstructing edge subsets in a single shot without any imposed ordering, resulting in a simpler and more flexible framework. Furthermore, BiGG is limited to unattributed graphs, whereas SparseDiff naturally supports diverse edge types and node features.

**HiGen**   HiGen (Karami, 2023) adopts a coarse-to-fine generation scheme, first producing community sub-graphs and then predicting inter-community edges. This hierarchical design assumes modular graph structure. SparseDiff makes no such assumptions, treating all edges uniformly through sparse diffusion.

**HGGT**   HGGT (Jang et al., 2023) encodes the adjacency matrix as a $K^2$-tree token sequence and processes it with a Transformer using tree-based positional encodings. This strategy embeds structural priors of using a predefined hierarchical data structure (the $K^2$-tree) into the generation process. In contrast, SparseDiff directly applies sparse diffusion over raw edge lists, avoiding dependence on any hierarchical encoding.

**SaGess**   SaGess (Limnios et al., 2023) is a discrete denoising diffusion model for graph generation that scales to large networks through a generalized divide-and-conquer strategy. It trains a base diffusion model (DiGress) on a collection of smaller subgraphs that collectively cover the original input graph. These sub-graphs are generated independently and subsequently assembled to form a complete synthetic network. In contrast to SparseDiff, which performs diffusion and precise denoising over the entire graph, SaGess achieves scalability by generating graph partitions independantly and merging them into a single graph.

**GraphLE**   GraphLE (Bergmeister et al., 2023) grows the graph through local expansion, starting from a seed node and incrementally adding nodes and edges via multiple diffusion passes. While it avoids full pairwise modeling, this approach is in practice resource-intensive and requires denoising for each expansion. SparseDiff instead performs the diffusion in a single process.

## E.2   Discrete Graph Diffusion

While several models have emerged using discrete diffusion on graphs (Vignac et al., 2023a; Haefeli et al., 2022), DiGress remains the most performant and is widely regarded as the benchmark.

Specifically, DiGress extends the discrete diffusion framework D3PM (Austin et al., 2021) to graph data by modeling both nodes and edges as categorical variables encoded in one-hot format. In contrast to SparseDiff,

which represents graphs using sparse triplets $(\boldsymbol{E}, \boldsymbol{X}, \boldsymbol{Y})$, DiGress encodes a graph $G = (\boldsymbol{X}, \boldsymbol{E})$ with $n$ nodes using a dense node feature tensor of shape $(n \times a)$ and an edge tensor of shape $(n \times n \times b)$, where the edge tensor includes an additional category for absent edges.

To process batches of graphs with variable sizes, DiGress zero-pads each graph to the dimensions $(bs, n_{\max}, a)$ for nodes and $(bs, n_{\max}, n_{\max}, b)$ for edges, where $bs$ is the batch size and $n_{\max}$ is the maximum number of nodes among all graphs in the batch. This padding strategy introduces substantial memory overhead and fails to scale for variable-sized graphs, motivating our use of a sparse representation to eliminate this limitation.

**Noise Model**   At each forward step $t$, DiGress applies categorical noise via transition matrices $\boldsymbol{Q}_X^t$, $\boldsymbol{Q}_E^t$, where $[\boldsymbol{Q}_X^t]_{ij} = P(x^t = j \mid x^{t-1} = i)$, $[\boldsymbol{Q}_E^t]_{ij} = P(e^t = j \mid e^{t-1} = i)$, defining

$$q(G^t \mid G^{t-1}) = \left(\boldsymbol{X}^{t-1} \boldsymbol{Q}_X^t, \ \boldsymbol{E}^{t-1} \boldsymbol{Q}_E^t\right), \quad q(G^t \mid G) = \left(\boldsymbol{X} \, \bar{\boldsymbol{Q}}_X^t, \ \boldsymbol{E} \, \bar{\boldsymbol{Q}}_E^t\right),$$

with $\bar{\boldsymbol{Q}}^t = \prod_{s=1}^{t} \boldsymbol{Q}^s$, as detailed in Section 3.1.1.

DiGress defines the forward process using $\bar{\boldsymbol{Q}}^t = \prod_{s=1}^{t} \boldsymbol{Q}^s$, and deviates from the uniform prior (Austin et al., 2021) by aligning the final state with the empirical marginal distribution of node and edge categories. This results in a better-initialized noisy graph, which leads to consistently improved performance.

In SparseDiff, we prove that the noisy graphs sampled along such trajectory have the same level of sparsity with the clean graph with high probability and also eliminate redundant computations present in dense diffusion schemes.

**Denoising Network**   The denoising network of DiGress is a permutation-equivariant full-attention graph transformer augmented with PNA (Corso et al., 2020) and FiLM (Perez et al., 2018) layers. Since it attends over all $n^2$ pairs, it captures long-range interactions but scales quadratically in memory.

In SparseDiff, we propose a message-passing based denoising network which adapts better to our sparse representation, governed by a sparse attention map that incorporates all existing edges along with a controllable number of random connections. This enables training on large graphs with no degradation in generation quality.

Under an independence assumption across node and edge entries, DiGress optimizes the cross-entropy loss

$$\mathcal{L} = \sum_{i=1}^{n} \mathrm{CE}\left(x_i, \hat{p}_i^{\boldsymbol{X}}\right) \ + \ \lambda \sum_{i,j=1}^{n} \mathrm{CE}\left(e_{ij}, \hat{p}_{ij}^{\boldsymbol{E}}\right),$$

where $\hat{p}^{\boldsymbol{X}}, \hat{p}^{\boldsymbol{E}}$ are the model's predicted category probabilities.

In SparseDiff, we perform the loss computation only on query edges randomly sampled from all node pairs, functionning as random batches of the edges considered in DiGress. To ensure stability across different sparsity levels, we also reweight the edge loss based on the sparsity parameter $\lambda$.

**Inference**   DiGress frames graph generation as a discrete diffusion process over nodes and edges independently, following the D3PM formalism. At each step $t$, the model observes a noisy graph $G_t = (\boldsymbol{X}^t, \boldsymbol{Y}^t)$ and samples $G_{t-1}$. For each scalar variable (i.e., a single node or edge) with one-hot state $x^t$, D3PM defines the exact posterior conditioned on the unknown clean label $x^0$ as:

$$q(x^{t-1} \mid x^t, x^0) = \frac{q(x^t \mid x^{t-1}) \, q(x^{t-1} \mid x^0)}{q(x^t \mid x^0)},$$

which, in matrix form, becomes:

$$q(x^{t-1} \mid x^t, x^0) = \mathrm{Cat}\left(x^{t-1}; \ p = \frac{x^t \mathbf{Q}^{t\top} \odot x^0 \bar{\mathbf{Q}}^{t-1}}{x^0 \bar{\mathbf{Q}}^t x^{t\top}}\right),$$

where $\mathbf{Q}^t$ is the one-step transition matrix, $\bar{\mathbf{Q}}^t = \prod_{s=1}^{t} \mathbf{Q}^s$, and "$\odot$" denotes element-wise multiplication.

DiGress begins its generation by sampling the number of nodes from the training graph size distribution. Then, at each diffusion step $t$, it replaces the intractable posterior weights $q(x_i^0 \mid G_t)$ and $q(e_{ij}^0 \mid G_t)$ with predictions from a denoising network, $\hat{p}_\theta(x_i^0 \mid G_t)$ and $\hat{p}_\theta(e_{ij}^0 \mid G_t)$. These are then used to compute the reverse transition:

$$p_\theta(x^{t-1} \mid G_t) = \sum_{x^0} q(x^{t-1} \mid x^t, x^0)\, \hat{p}_\theta(x^0 \mid G_t).$$

Nodes and edges are evaluated independently, and each categorical variable is sampled accordingly. Iterating this process from $t = T$ to 1 produces the final generated graph.

In SparseDiff, we modify this sampling process to align with the sparse attention mechanism used during training, ensuring both structural consistency and computational efficiency at inference time.

**Summary**   In summary, DiGress explicitly addresses the discrete and permutation-equivariant nature of graph data, outperforming existing one-shot generation methods in molecular graph synthesis. However, its reliance on dense matrix representations limits scalability to graphs.

## F   Visualization

(a) Training graphs.

(b) Generated graphs.

Figure 8: Visualization for QM9 dataset with implicit hydrogens.

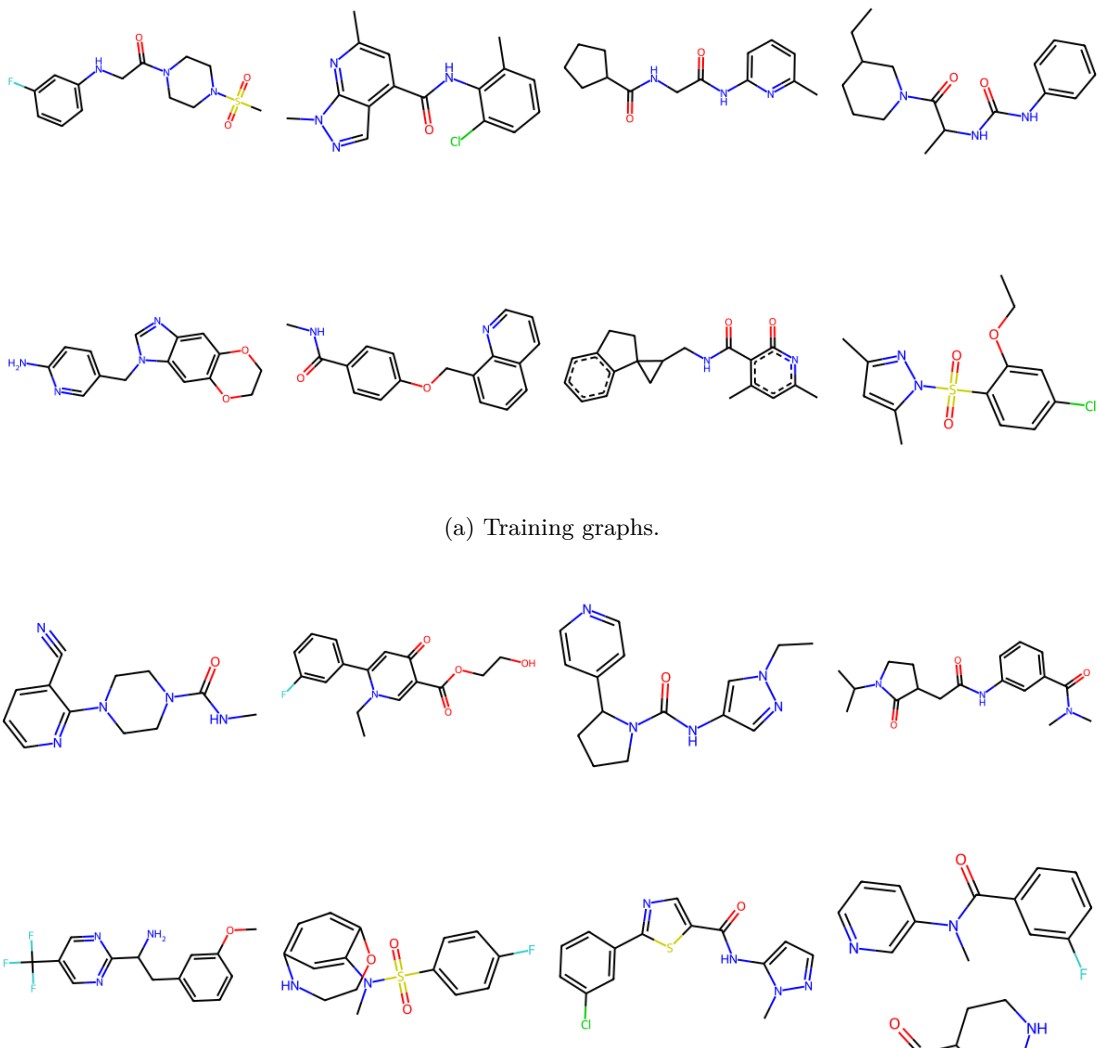

(a) Training graphs.

(b) Generated graphs.

Figure 10: Visualization for MOSES dataset.

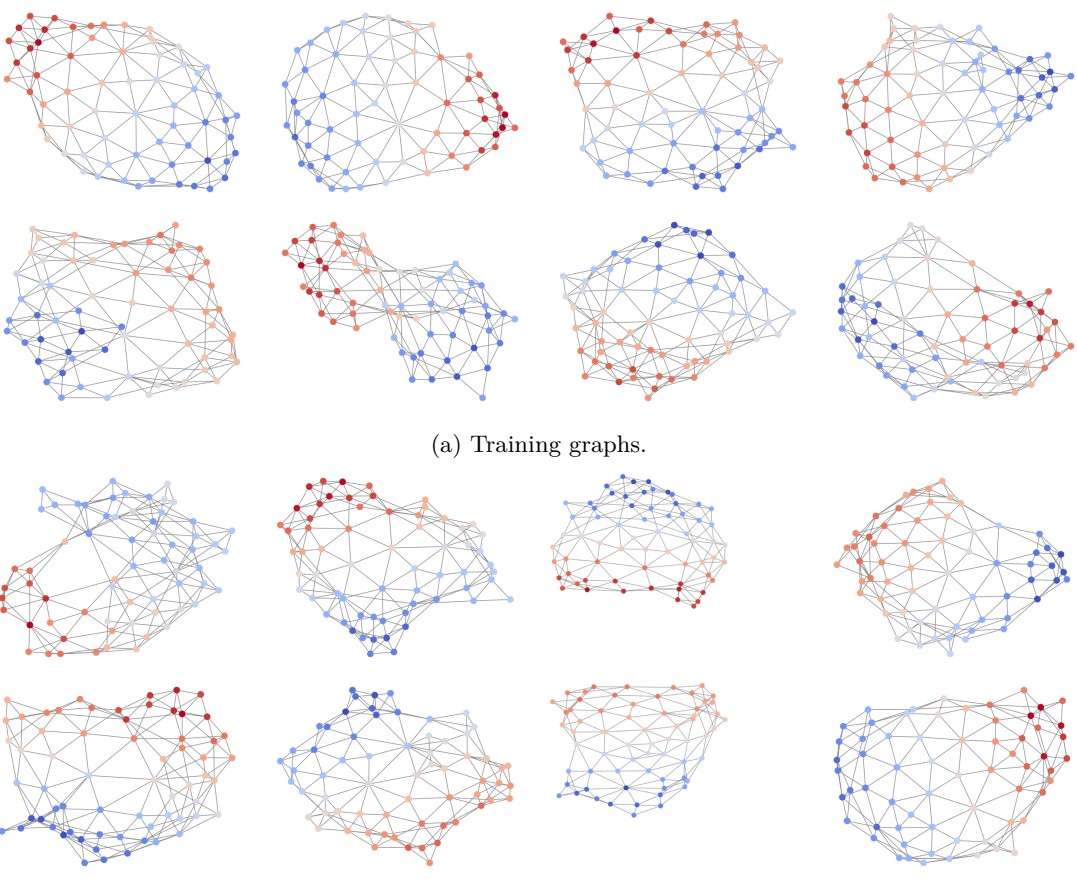

(a) Training graphs.

(b) Generated graphs.

Figure 11: Visualization for Planar dataset.

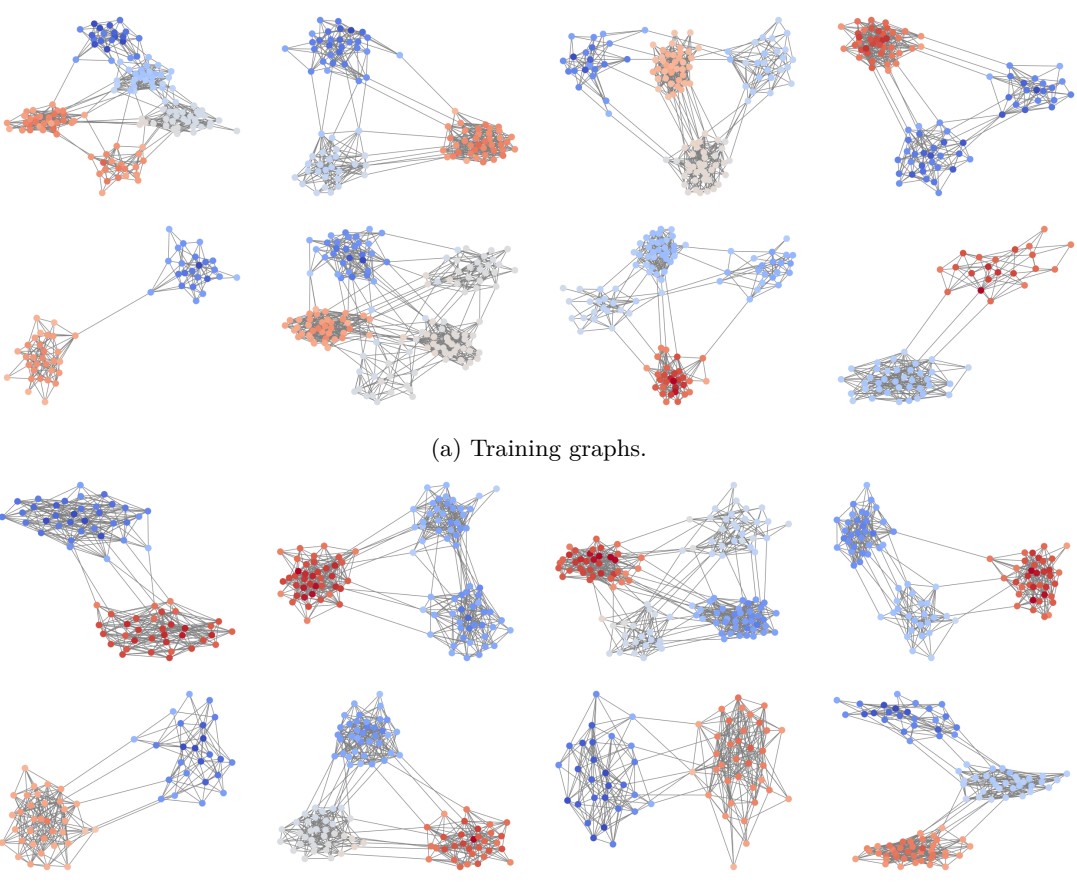

(a) Training graphs.

(b) Generated graphs.

Figure 12: Visualization for SBM dataset.

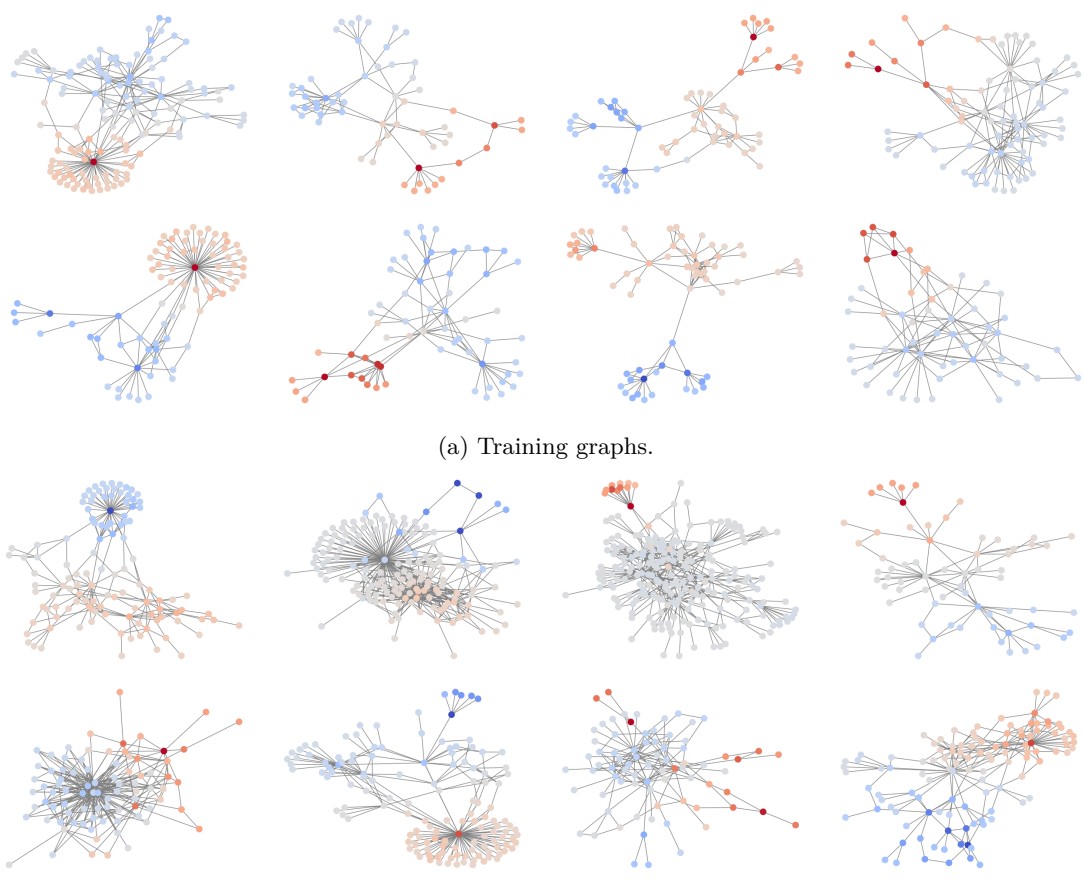

(a) Training graphs.

(b) Generated graphs.

Figure 13: Visualization for Ego dataset.

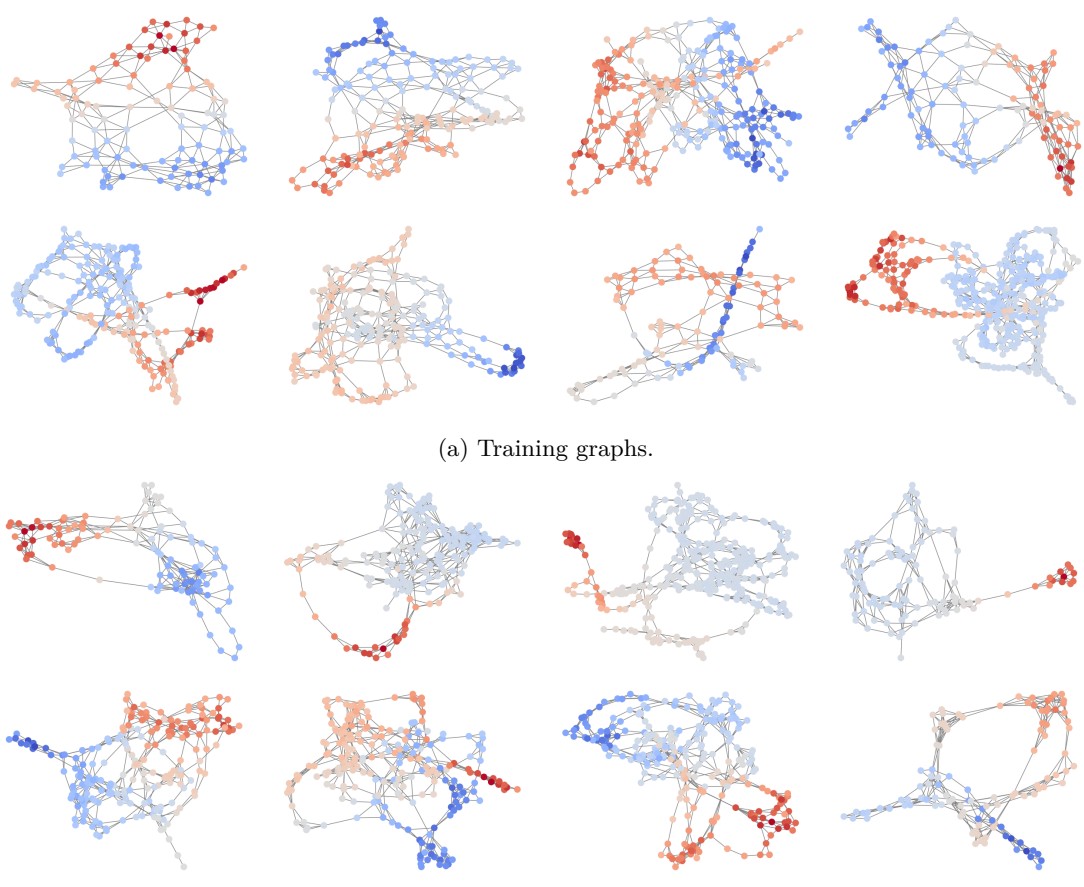

(a) Training graphs.

(b) Generated graphs.

Figure 14: Visualization for Protein dataset.

