# OpenReview forum: "SparseDiff: Sparse Discrete Diffusion for Scalable Graph Generation"
_TMLR — Accepted by TMLR_

### Review · Reviewer_tUWL · 2024-11-21

**Summary Of Contributions:**

This paper proposed SparseDiff, a novel diffusion framework that leverages graph sparsity to efficiently scale graph generative models by reducing complexity in noising, training, and inference, achieving state-of-the-art performance, robust scalability, and faster convergence on large-scale datasets.

**Audience:**

Yes

**Claims And Evidence:**

Yes

**Requested Changes:**

- It would be more convincing to add some analysis of space complexity.
- More formal definitions or symbolized equations should be given when introducing the details of the method.
- Additional experiments should be put into the main pages to better show the proposed method's effectiveness.
- More theoretical analysis should be added.

**Strengths And Weaknesses:**

Strengths:
1. The motivation is clear. It holds great significance that leveraging the sparsity in large graphs to enable efficient sparse modelling without
sacrificing generation quality.
2. The method is clear and easy to follow. Fig.1 gives a very good illustration of the whole method.
3. The proposed method has been tested on diverse graph datasets to demonstrate its scalability and versatility.

Weaknesses:
1. While the author shows the time efficiency in the experiments, the analysis of space complexity is missing.
2. Besides description, a more formal definition or symbolized equation should be given, especially in Sec.3.1.2 and Sec.3.2.1.
3. Lack of Thorough Theoretical Analysis. It is essential to provide a more rigorous theoretical foundation to support the claim of improved efficiency.

---

> ### Author Response · Authors · 2025-03-09
>
> We appreciate your recognition of our work and are glad that our motivation, methodology, illustration, and empirical validation were well-received. Your feedback has been very constructive, and we address your concerns as follows:
>
> **W1** Thank you for suggesting adding this important aspect.
> We have added a quantitative comparison in Figure 5 (page 11), illustrating the memory consumption of DiGress and SparseDiff under the same batch size with varying sparsity parameters. Specifically, we demonstrate an approximate linear relationship between the sparsity parameter $\lambda$ and actual space complexity on the QM9 and SBM datasets, highlighting the effectiveness of our sparsification technique。
>
> For example, on the SBM dataset, training with $\lambda=0.25$ reduces memory usage to 31.8% of that required by DiGress. At $\lambda=1.0$, where sparsity is maximized, our method also improves efficiency by avoiding the overhead of dense models, which enforce a fixed size of $(n_{\max}, n_{\max})$ in batched computations, where $n_{\max}$ denotes the largest node count within the batch.
>
> On the QM9 dataset, where graph sizes are smaller and more uniform, our method maintains an approximately linear reduction in memory usage. At $\lambda=1.0$, SparseDiff exhibits a slightly higher space complexity due to additional indexing operations inherent to message-passing mechanisms and sparse graph representations, while still effectively reducing overall resource consumption with smaller $\lambda$.
>
> **W2** Thank you for your suggestion to improve clarity. We have revised the manuscript as follows: In Section 3.1.2, we explicitly define existing edges as $E$ and non-existing edges as $E_{nq}$, ensuring clearer notation. We also refine the three-step process with more detailed mathematical formulations. In Section 3.2.1, we formally define $E_m$ and $E_q$ for message passing, complementing the visualization in Figure 3. These updates further formalize the notations of our manuscript.
>
> **W3** To provide deeper theoretical insights into the space complexity of our framework, we have added a subsection in Appendix A.1 with a detailed space complexity analysis of SparseDiff.
>
> In response to the requested changes, we have made the following modifications to our manuscript.
> * Space Complexity Analysis: see W1
> * Formel Definitions: see W2
> * Additional Experiments: We incorporated the important ablation study on the sparsity parameter and added a visualization comparing the space complexity of SparseDiff and DiGress into the main paper. Additionally, we included the MOSES molecular dataset to further demonstrate our performance.
> * Theoretical Analysis: see W3
>
> We appreciate the reviewer's valuable feedback, and we look forward to meeting the requirements in the revised version.

---

### Review · Reviewer_vE9L · 2024-11-28

**Summary Of Contributions:**

The authors propose SparseDiff, a novel graph diffusion model focusing on exploiting the sparsity of graph-structured data. One issue of dense graph generative models is having to deal with adjacency matrices, which scale quadratically with the number of nodes. The diffusion and denoising processes of SparseDiff operate instead on the edge list, whose size depends on the sparsity of graphs. As is done in DiGress [1], the diffusion process randomly flips the labels of nodes and edges, but differently from it, only existing edges are modeled explicitly. Non-existing edges go through 3 stages of sampling: the number of new edges,  their position, and their label. To keep the graph sparse during diffusion, the authors opt for the marginal transition process and accurately analyze the evolution of the number of edges. During training, the denoising model is trained to classify the correct existing edges but also addresses currently non-existing edges through random attention. At inference, all node pairs must be considered. The authors propose to do so by partitioning the square adjacency matrix into equally sized sets of edges, and iteratively sampling their new label for the next step using the above-mentioned denoising model. SparseDiff is evaluated on many small, medium, and large datasets against several baselines, from dense models to hierarchical and sparse models. Results show promising results in the direction of sparse conversion of dense models, such as SparseDiff.

**Audience:**

Yes

**Claims And Evidence:**

Yes

**Requested Changes:**

As discussed in the weaknesses, critical changes comprise:
- a deeper comparison between DiGress and SparseDiff, with a stronger motivation for the better results achieved.
- mentioned bold statements being revisited, or better motivated.

Changes that would likely enhance the value of the paper:
- describing, if possible, a recipe for applying the same sparsification to other dense models
- having an evaluation of the memory consumption of the model against others

Possible typos on page 3, section 2.1: notation is inconsistent for $x$ (clean data) and $z_t$ (corrupted data). For example $x_t$ should be $z^t$, and $\epsilon=z^t-x$.


[1] Vignac, Clement, Igor Krawczuk, Antoine Siraudin, Bohan Wang, Volkan Cevher, and Pascal Frossard. "DiGress: Discrete Denoising diffusion for graph generation." In _The Eleventh International Conference on Learning Representations_ (2023).

[2] Chen, Xiaohui, Jiaxing He, Xu Han, and Li-Ping Liu. "Efficient and degree-guided graph generation via discrete diffusion modeling." In _Proceedings of the 40th International Conference on Machine Learning_, pp. 4585-4610. 2023.

**Strengths And Weaknesses:**

Strengths:
1. The paper is generally well-written and well-structured. I found it understandable, and the figures are appropriate and clarify some key points, for example, Figures 2 and 3. I appreciated the thoroughness of the experimental evaluation.
2. Some interesting key points are addressed in this work. One such example is the unfeasibility of applying dense models to larger graphs, but still having the need to model all node interactions, which is done in this work in a sparse way through random attention. Another instance is being able to tune the sparsification of the model, starting from a reference dense model. This work clearly takes inspiration from DiGress [1], for which SparseDiff can be seen as its sparse version. This reasoning could be extended to other dense models.
3. Results show that sparsification does not hinder the expressivity of a dense model, but sometimes even elevates it. Additionally, better memory management helps in applying the model to much larger graphs.

Weaknesses:
1. Although the model seems novel as a whole, upon a deeper inspection, it is actually using the DiGress [1] diffusion and denoising processes. The change is in the representation used, which only materializes edges with a label and implicitly samples non-existing edges. Technically, the two models's diffusion processes should align perfectly, and the same is true for the denoising process, depending on the parameter $\lambda$. Although this is not a weakness in itself, I would have appreciated this parallel being highlighted, and possibly having some guidelines for applying the same recipe to other dense models, if the authors find it possible.
2. As explained in the previous point, intuitively, SparseDiff approximates DiGress, although with a different architecture. For this reason, I find it odd that SparseDiff can outperform the latter. One reason I can find is the change in architecture, and the fact that the inductive bias on sparsity is beneficial. I would like the authors to elaborate on this point.
3. It seems there are a few bold statements regarding the line of hierarchical models. One such instance is "While these approaches are designed specifically for large graphs, they rely on additional assumptions such as cluster structures or dependencies on node degrees which can hinder their generalizability."  Although it is true that these models exploit some properties of graphs, the motivation the authors provide is weak. Experimentally, I did not see this drop in generalization, and intuitively, these models softly follow these directions. For example, in EDGE [2] it is true that not all node degree sequences are feasible, but degrees are given as a conditioning feature to the model, not enforced through hard constraints.
4. I would have appreciated a quantitative analysis of different memory consumptions, which should highlight the sparsification technique's impact, other than generation quality and training time.

---

> ### Author Response · Authors · 2025-03-09
>
> Thank you for recognizing our work. We are glad that the clarity of our writing, the effectiveness of our figures, and the contributions of controllable memory usage and the high performance of our proposed model. Your valuable insights have helped us refine the manuscript, and the concerns you highlighted are addressed below:
>
> **W1** Thank you for your insightful feedback regarding the relationship between SparseDiff and DiGress [1]. SparseDiff indeed follows the discrete diffusion framework of DiGress. Beyond that, its sparsification enables greater flexibility in memory usage while maintaining alignment with DiGress in diffusion and denoising, depending on the parameter $\lambda$.
> In response to your suggestion, we have explicitly highlighted this parallel in the manuscript and provided guidelines on how SparseDiff’s sparsification strategy can be applied to other dense models, as added in the end of the introduction Section, shown as follows:
>
>     SparseDiff is implemented within the discrete diffusion framework, specifically following the setup of \citet{vignac2022digress}. Precisely, its noise model is particularly well-suited for scenarios where non-existent edges dominate, such as as marginal trajectories in SparseDiff or absorbing trajectories that converge to empty graphs. Additionally, the sparse attention mechanism and iterative sampling are broadly applicable to both diffusion or flow-based generative models, as they focus solely on sparsifying the attention map and the corresponding adjacency matrix. As a remark, while SparseDiff improves diffusion and flow models with sparse noisy graphs, it provides no benefit for continuous models that do not preserve sparsity, as they must be modeled using a dense adjacency matrix.
>
> **W2** Thank you for your question regarding SparseDiff’s superior performance over DiGress.
> In our experiments, we evaluate both small and large datasets. For smaller datasets, our goal is to match DiGress's performance, which we successfully achieve on Moses, QM9, SBM, and Planar.
> For larger datasets, SparseDiff’s advantage comes from its training efficiency, which allows us to train the model to convergence. Due to its sparse modeling, SparseDiff supports significantly larger batch sizes under the same computational constraints. For example, on the SBM dataset, DiGress takes up to 7 days to train, while SparseDiff converges in only 3 days by leveraging a larger batch size on an equivalent GPU. Moreover, on datasets with more than 1,000 nodes, DiGress runs out of memory even with a batch size of 1, whereas SparseDiff remains trainable.
>
> One reason large graphs are especially challenging to model is the lack of a fixed spatial structure. Unlike images, where nearby pixels are correlated and can be processed efficiently using convolutional architectures, graphs are permutation-invariant, meaning node positions carry no inherent meaning. As a result, relationships between distant nodes must be learned independently, making large-scale prediction more difficult. SparseDiff mitigates this challenge with its efficient sparse modeling, allowing it to scale where dense model struggles.
>
> **W3** Thank you for your insightful feedback. We recognize that our initial statement may have been too assertive and have revised it for greater clarity:
>
>     Overall, scalable generation models often introduce additional dependencies on node orderings, incorporate assumptions about data distributions, or leverage specific graph properties, while latent graph diffusion faces challenges for graph matching. In contrast, the SparseDiff model described in the next section aims at making no assumption besides graph sparsity, offering a more general and streamlined framework for graph generation, while preserving competitive performance across a wide range of graphs with different sizes.
> Regarding EDGE, while degree information is applied as a conditioning feature rather than a strict constraint, it may still introduce misalignment between training and sampling. Specifically, the sampling process can produce degree distributions that are infeasible for undirected graphs, which do not occur during training. We further motivate our approach from the performance perspective. We have updated our statement accordingly:
>
>     While this factorization is universally applicable, the conditional distribution $p_\theta(\mA | \vd^0)$ may occasionally lead to degree distributions that are not feasible for undirected graphs during sampling, introducing a slight misalignment between training and generation. EDGE’s performance also falls behind than SparseDiff on both small and large graphs.

---

> > ### Author Response · Authors · 2025-03-09
> >
> > **W4** We have added a quantitative comparison in Figure 5 (page 11), illustrating the memory consumption of DiGress and SparseDiff under the same batch size with varying sparsity parameters. Specifically, we demonstrate an approximate linear relationship between the sparsity parameter $\lambda$ and actual space complexity on the QM9 and SBM datasets, highlighting the effectiveness of our sparsification technique.
> >
> > For example, on the SBM dataset, training with $\lambda=0.25$ reduces memory usage to 31.8% of that required by DiGress. At $\lambda=1.0$, where sparsity is maximized, our method also improves efficiency by avoiding the overhead of dense models, which enforce a fixed size of $(n_{\max}, n_{\max})$ for batched computations.
> >
> > On the QM9 dataset, where graph sizes are smaller and more uniform, our method maintains an approximately linear reduction in memory usage. At $\lambda=1.0$, SparseDiff exhibits a slightly higher space complexity due to additional indexing operations inherent to message-passing mechanisms and sparse graph representations, while still effectively reducing overall resource consumption with smaller $\lambda$.
> >
> > According to the requested changes, we carefully corrected the typos and made the following improvements to enhance the manuscript’s quality:
> >
> > * Comparison of SparseDiff and DiGress: see W1
> > * Stronger motivation for the superior experimental results: see W2
> > * Statements to be revisited: see W3
> > * Recipe to adapt SparseDiff: see W1
> > * Memory consumption: see W5
> >
> > We appreciate the reviewer's valuable feedback, and we look forward to meeting the requirements in the revised version.
> >
> > [1] DiGress: Discrete Denoising diffusion for graph generation, ICLR 2023, Vignac et al.

---

> > > ### Comment · Reviewer_vE9L · 2025-04-28
> > > **Answer to Authors**
> > >
> > > Dear authors,
> > >
> > > Thank you for the clarifications and improvements to the paper. Let me quickly point out a typo: "Impact of the **Sparseity** Parameter" in the paragraph under Figure 5, page 11.
> > >
> > > Given the new theoretical insights and analyses on efficiency, and the abundance of empirical proof, I found all my questions answered and would advise acceptance. The paper offers a new way to scale graph generation to very large graphs, an aspect of utmost importance in the field.

---

> > > > ### Author Response · Authors · 2025-05-03
> > > >
> > > > We are glad to have addressed the questions and appreciate the reviewer’s suggestions, which significantly improved our writing.
> > > >
> > > > We corrected the typo and cross-checked the revised content.
> > > >
> > > > We thank the reviewer again for their positive assessment and recommendation for acceptance.

---

### Review · Reviewer_x8GY · 2025-04-14

**Summary Of Contributions:**

The paper proposes a graph diffusion-based approach by leveraging the sparsity of large graphs and building on top of the DIGRESS framework [1].

NOTE: Overall, I feel that the reader should be extremely familiar with the diffusion on graphs literature to understand the implementation details and rigor of the paper. It would be great if the authors can address the feedback below and make their work more accessible.

[1] Vignac, Clement, et al. "Digress: Discrete denoising diffusion for graph generation." arXiv preprint arXiv:2209.14734 (2022).

**Audience:**

Yes

**Claims And Evidence:**

Yes

**Requested Changes:**

1. For each of the methods considered as a baseline in this work, can the authors briefly add some information about the technical details of the algorithms? In particular, where does SparseDiff differ from these previous works so that the readers can clearly understand the pain points that this paper addresses?

2. It would be great to have a notations section in the paper to clearly explain what each symbol means in the paper. For example, I find it difficult to interpret Eq 3 even though it is a CE loss calculation. What computation steps are required to obtain $\hat{P}^G_q$ ? I observed that step 6 in algorithm 1 can result in this value but what exactly in $\phi_{\theta}$ in step 6 ?

3. Can the authors clarify how their sparse computation and the iterative inference process are related in terms of memory savings?

4. Can the authors provide a breakdown of the memory savings from the three phases of the denoising process? Where exactly is the memory saving coming from?

**Strengths And Weaknesses:**

**Strengths:** The idea of leveraging graph sparsity for diffusion-based generation seems interesting and practical. The paper also builds on existing work and is well-motivated.

**Weaknesses:** I believe that the presentation of ideas needs to be improved. Especially:

1. The technical contributions that are unique to this work are unclear. In particular, it is not straightforward to easily understand the contributions based on the current organization of content.

2. The technical detail that requires further clarification and is contrary to the claims is the usage of $E_{ne}$ to sample edges in Section 3.1.2. Essentially, one can think of an adjacency matrix as a combination of $E$ + $E_{ne}$ which requires $O(n^2)$ space complexity. The authors state that they do not materialize such a matrix and rely on the sparsity of the graph (i.e, assume that the set of edges $E$ is small enough) for their computations. This is a bit confusing since I do not understand how "Step 2: Sampling positions" is possible without materializing $E_{ne}$.  This whole section was confusing to me.

3. The authors have not presented any details about the hyper-parameters used for training. Also, the details about the number of layers, hidden dimensions, etc used for the GNN architecture are missing in Appendix B. Furthermore, the details about how the baseline numbers were measured is also missing.

I am willing to reconsider my opinion about "Claims and Evidence" if the authors can address the above comments.

---

> ### Author Response · Authors · 2025-04-23
>
> We thank the reviewer for positively assessing our work and for the constructive suggestions. We respond to the raised weaknesses below:
>
> **Technical contributions** As the reviewer correctly understood, our work leverages graph sparsity by employing a sparse edge list representation to improve the scalability of graph diffusion models. In short, our main technical contributions are as follows.
>
> 1. We contribute first by observing that real-world graphs are typically highly sparse, which motivates the use of sparse representations based on edge triplets $(E, X, Y)$ rather than the dense formats adopted by previous graph diffusion models. This representation aligns with graph structure and scales more efficiently with variable graph sizes (see Appendix E.2 for details).
> 2. Building on this, we introduce a tailored noising trajectory that preserves sparsity with high probability, ensuring that sparsity assumption holds throughout the entire diffusion and denoising process. We also improve the computation of the noising process to be more space efficient.
> 3. We then design a space-efficient denoising network using convolutional attention layers adapted to our sparse graph representation. Its complexity depends on the entries of a sparse attention map covering all edges and a tunable number of random connections, enabling training on large graphs without compromising quality.
> 4. We modify the sampling process to be iterative, aligning it with the sparse attention mechanism used during training. At each step, a non-repetitive subset of edges is generated, progressively constructing a complete attention map. This design enables space-efficient generation of large graphs.
>
> These design choices are novel compared to existing approaches such as Digress and allow SparseDiff to outperform them. Following the reviewer suggestion, we now explicitly list our contributions in Section 1 Introduction.
>
> Our experiments further support the technical contribution by showing that SparseDiff scales to graphs with up to around 2,500 nodes, achieving state-of-the-art performance across all scales. In contrast, the performance of DiGress degrades on graphs with over 200 nodes.
>
> **Clarification on position sampling** We thank the reviewer for the question. The reviewer is indeed correct with the materialization of $E_{nq}$ for sampling positions described in Section 3.1.2.
>
> We clarify that we can not avoid materializing $E_{nq}$. Specifically, in Step 2 of Section 3.1.2, sampling edge positions contains operations over the complement of the current edge set, i.e., over $O(n^2 - m)$ candidate positions. We provide a detailed space complexity analysis for Section 3.1.2 in Appendix A.1.1.
>
> Nevertheless, our method explicitly exploits sparsity by operating directly on edge lists, which maintains the same mathematical formulation with dense models while significantly reducing memory overhead. In Appendix A.1.1, we now provide a concrete example demonstrating a 3X reduction in space complexity compared to prior dense approaches. Notably, while each component of SparseDiff is designed for efficiency, the main computational bottleneck lies in training the denoising network, and the cost of the noise model is negligible. This bottleneck can be effectively managed through our proposed tunable parameter $\lambda$, as illustrated in Fig.5.
>
> Upon this, we provide a new revision with a more accurate analysis, aligned with the implementation.
>
> We hope this clarification resolves the confusion.
>
> **Hyperparameters and architecture details** We have updated Section 4 and Appendix D.6 to explicitly report all hyperparameters and training details. To make the comparison fair, SparseDiff adopts the configuration of DiGress without additional tuning. The only introduced hyperparameter is the sparsity controller $\lambda$, which is selected based on graph size (Table 9, Appendix C.2). All implementation and technical details are also included in the anonymous codebase.
>
> Concerning the architecture, we follow the previous work DiGress and adopt the graph transformer with normalization, feed-forward, and attention layers. We incorporate PNA pooling and FiLM conditioning to improve predictive accuracy and computational efficiency. However, we use sparse convolutional attention layers to replace its dense attention layers to effectively handle sparse inputs. Detailed parameters (e.g., number of layers, hidden dimensions) are now explicitly included in the updated Appendix D.6.
>
> **Baselines** All baseline results follow configurations from their original papers to ensure a fair comparison. For datasets where DiGress checkpoints were unavailable (SBM, Planar, QM9), we reimplemented and re-evaluated DiGress under identical settings and metrics as SparseDiff. We included this information in Section 4.

---

> > ### Author Response · Authors · 2025-04-23
> >
> > # Requested changes
> >
> > **C1 Baselines** We thank the reviewer’s suggestion and agree that having an explicit comparison between SparseDiff and prior methods can clearly demonstrate SparseDiff’s advantage and motivation.
> >
> > * **Scalable methods:** We expanded our discussion on scalable baseline methods in Appendix E.1 in the revised version, providing more precise descriptions to complete our previous discussion in Section 1 Introduction and in Section 2 Related Work.
> >
> > * **Accessibility:** We additionally provide a focused overview of DiGress in Appendix E.2, offering the necessary background for readers to understand the improvements introduced by SparseDiff.
> >
> > **C2 Notation** We thank the reviewer for their feedback. We have defined $\theta$ in Section 2.1 and $\phi$ in Section 3.3 at their first appearance in the main text. As suggested by the reviewer, to eliminate any ambiguity, we added a dedicated notation table (Table 9, Appendix A, Page 17) summarizing all symbols used throughout the paper in the revision. We have also revised the description around Equation 3 and Algorithm 1 to explicitly link $\hat{P}^G_q$ to Step 6, ensuring full traceability of the CE loss computation.
> >
> > **C3 Sparse computation and memory complexity** Intuitively, SparseDiff avoids constructing or operating over the full $n^2$ adjacency matrix. Instead, it retains only the existing edges and samples $\lambda n^2$ query edges to approximate full attention for message passing. Since the space complexity of our sparse convolutional attention layers scales linearly with the number of message-passing edges, now controlled directly by $\lambda$. Please refer to **C4 Component-wise memory analysis** for quantitative component-wise memory analysis.
> >
> > **C4 Component-wise memory analysis** We thank the reviewer for their question. We designed all components of SparseDiff to be as space-efficient as possible. Among them, the denoising network is the primary memory bottleneck during training. In contrast to dense baselines such as DiGress, which scale as $O(n^2)$ with a large constant $d_e \cdot L$ with $L$ as the number of layers and $d_e$ as the dimension of edge activations, typically set to 64 or 128, SparseDiff constrains memory usage by operating on a sparse set of sampled edges scaling with $O(\lambda n^2)$. This is further supported by Fig.5 in the revised manuscript, which shows that GPU memory usage scales approximately linearly with $\lambda$. Additionally, our experiments demonstrate that the model can handle graphs with up to approximately 2500 nodes.
> >
> >  In the revision, we included in Appendix A.1 a detailed breakdown of memory savings per component of SparseDiff. Briefly, we conclude:
> >
> > 1. **Efficient Noise Model:** SparseDiff reduces noise application complexity to $O(2n^2)$ instead of $O(3bn^2),\ b\geq 2$ by avoiding redundant computations and leveraging sparsity in non-existing edge processing.
> > 2. **Efficient Denoising Neural Network:** SparseDiff’s denoising network operates over a sparse attention map, scaling as $O(\lambda n^2)$, unlike dense baselines (e.g., DiGress), which process a full attention map for all $n^2$ node pairs.
> > 3. **Iterative Inference:** Inference reuses sparsely sampled edges from training, keeping its space complexity similarly scaling with $O(\lambda n^2)$.
> >
> > We thank the reviewer again for their comments and believe the presentation of the paper has significantly improved following their feedback. We hope our revision resolves the concern. We remain open to any clarification.

---

> > > ### Comment · Reviewer_x8GY · 2025-04-29
> > >
> > > Thanks for the response. The updated version looks much better.
> > >
> > > nit: In Appendix E.2 first paragraph, there is no need to mention "We provide a detailed discussion of DiGress in Appendix E.2".
> > >
> > > Please cross-check for such minor issues again.

---

> > > > ### Author Response · Authors · 2025-05-03
> > > >
> > > > We are glad that the revised version resolved the earlier concerns.
> > > >
> > > > We corrected the phrasing in Appendix E.2, and cross-checked the revised content.
> > > >
> > > > We thank the reviewer again for their positive feedback and constructive suggestions.

---

### Decision · Action_Editor_qE9V · 2025-06-03

**Recommendation:** Accept as is

**Audience:**

Yes

**Audience Explanation:**

Generation of graphs, especially by diffusion based approaches, is a relatively hot topic, in particular when considering drug and new material design.

**Claims And Evidence:**

Yes

**Claims Explanation:**

In order to cope with the generation of very large graphs, the paper shows how to scale one-shot graph generative models (DIGRESS) to work on sparse adjacency matrices. The claim on the efficacy of the proposed approach are supported by a large body of empirical evaluations. All other doubts about not supported claims have been clarified to reviewers during authors' rebuttal. The current version of the paper makes supported claims.